# PRIVACY BEYOND PIXELS: LATENT ANONYMIZATION FOR PRIVACY-PRESERVING VIDEO UNDERSTANDING

**Joseph Fioresi[1], Ishan Rajendrakumar Dave[2], Mubarak Shah[1]**
[1]Institute of Artificial Intelligence, University of Central Florida [2]Adobe
`joseph.fioresi@ucf.edu, idave@adobe.com, shah@crcv.ucf.edu`

## ABSTRACT

We introduce a novel formulation of visual privacy preservation for video foundation models that operates entirely in the latent space. While spatio-temporal features learned by foundation models have deepened general understanding of video content, sharing or storing these extracted visual features for downstream tasks inadvertently reveals sensitive personal information like skin color, gender, or clothing. Current privacy preservation methods focus on input-pixel-level anonymization, which requires retraining the entire utility video model and results in task-specific anonymization, making them unsuitable for recent video foundational models. To address these challenges, we introduce a lightweight Anonymizing Adapter Module (AAM) that removes private information from video features while retaining general task utility. AAM can be applied in a plug-and-play fashion to frozen video encoders, minimizing the computational burden of finetuning and re-extracting features. Our framework employs three newly designed training objectives: (1) a clip-level self-supervised privacy objective to reduce mutual information between static clips, (2) a co-training objective to retain utility across seen tasks, and (3) a latent consistency loss for generalization on unseen tasks. Our extensive evaluations demonstrate a significant **35%** reduction in privacy leakage while maintaining near-baseline utility performance across various downstream tasks: Action Recognition (Kinetics400, UCF101, HMDB51), Temporal Action Detection (THUMOS14), and Anomaly Detection (UCF-Crime). We also provide an analysis on anonymization for sensitive *temporal* attribute recognition. Additionally, we propose new protocols for assessing gender bias in action recognition models, showing that our method effectively mitigates such biases and promotes more equitable video understanding.

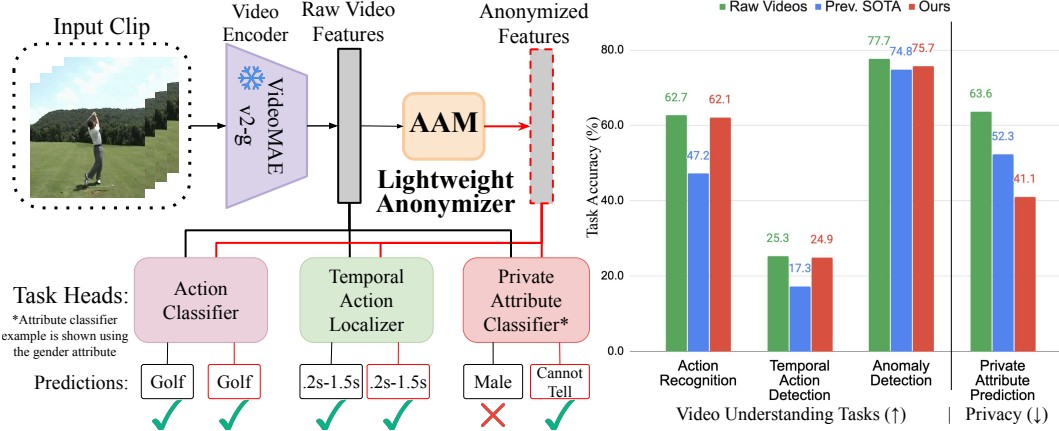

Figure 1: Our proposed latent anonymization setup (red) utilizes large pretrained video encoders, applying a lightweight anonymizer that maintains performance on *multiple* video understanding tasks while strongly reducing performance on private attribute prediction tasks (right).

# 1 INTRODUCTION

Video foundation models (VFMs) capable of addressing multiple video understanding tasks Wang et al. (2022; 2023); Bardes et al. (2024b); Dave et al. (2024) have been developed, but due to powerful modeling capabilities, the visual features extracted by these models reveal sensitive private information such as skin color, clothing, or gender (Figure 1). The high quality spatio-temporal features have enabled usage in real-world scenarios such as patient monitoring, sports analytics, robotics, and surveillance, where it is common practice to extract visual features, store them, and utilize them across multiple tasks. However, an attacker can use a classifier on these features to disclose such private attributes Fioresi et al. (2023), making it unsafe to store or share the visual features directly. This raises the question of how to protect an individual's private information in the feature space while maintaining the powerful video understanding capabilities of the VFMs.

Prior privacy-preserving methods Wu et al. (2020); Dave et al. (2022b); Li et al. (2023b); Fioresi et al. (2023) anonymize video models at the expense of utility. These works have primarily focused on input-level (i.e., pixel-level) privacy, which has limited applicability for two main reasons: (1) Since it alters the input, it requires retraining the utility model on the transformed data, which is impractical for large pretrained models trained on millions of videos with specific training recipes. (2) Existing methods have only proven efficacy on single downstream tasks. For instance, SPAct Dave et al. (2022b) is suitable only for action recognition, whereas TeD-SPAD Fioresi et al. (2023) is limited to anomaly detection. Consequently, the existing pixel-level anonymization formulation is not suitable to adopt to new developments in utility video foundation models.

To overcome these limitations, we propose a novel approach for privacy preservation in the latent feature space, demonstrated in Fig. 1. This design is well-suited for practical applications where features are stored for video search or analysis tasks. We call our method **SPLAVU**: **S**elf-supervised **P**rivacy-preservation via **L**atent **A**nonymization for general **V**ideo **U**nderstanding. SPLAVU is the first generalizable method to preserve privacy across diverse tasks *without requiring task-specific fine-tuning*, natively supporting tasks like action recognition, temporal action localization, and video anomaly detection while integrating seamlessly with various video foundation models.

To anonymize the utility model latent space, we introduce a lightweight, learnable Anonymizing Adapter Module (AAM) on a frozen VFM. Importantly, in contrast to prior methods Wu et al. (2020); Dave et al. (2022b); Fioresi et al. (2023), AAM is applied to temporal *clip-level* features instead of individual frames, allowing the anonymizer to communicate across the temporal dimension, more naturally aligning with video tasks. Our training framework employs three key objectives: (1) a clip-level self-supervised privacy preservation objective to minimize mutual information between two static clips, (2) a co-training utility objective to maintain performance across predefined tasks, and (3) a latent consistency loss to ensure generalization on unseen tasks. SPLAVU integrates seamlessly with multiple state-of-the-art methods for downstream tasks, significantly outperforming previous privacy-preserving methods. Additionally, SPLAVU is data-efficient; even when trained on a small dataset like HMDB51 Kuehne et al. (2011), it generalizes effectively without compromising the privacy-utility tradeoff. Under the right conditions, it can even defend against recognition of *motion-based sensitive attributes* such as gait. Beyond privacy protection, our work addresses the emerging issue of human-attribute bias in video understanding. For instance, models may exhibit gender biases by associating certain actions with specific genders based on stereotypes. For the first time, we introduce protocols to evaluate and mitigate gender bias in action recognition models.

Our contributions can be summarized as follows:

- We introduce a novel formulation of privacy preservation for general video understanding applications by anonymizing the latent embedding space.
- To enable latent anonymization, we propose a clip-level self-supervised privacy budget objective along with a latent consistency loss to preserve the utility generalization capability.
- Our method is the first to demonstrate privacy preservation across *multiple* downstream tasks, achieving a notable decrease in privacy leakage of over **35%** while preserving performance within **1-2%** across each utility task. Extensive ablation studies demonstrate the data efficiency of SPLAVU and its applicability across various video backbones.
- We additionally propose new protocols to assess gender bias in existing action recognition models and demonstrate that our method effectively mitigates this bias.

## 2 RELATED WORK

Video understanding spans tasks like action recognition, temporal action localization, and weakly-supervised anomaly detection. Various datasets have been introduced Carreira & Zisserman (2017); Diba et al. (2020); Goyal et al. (2017b); Zhao et al. (2019), and recent advancements include self-supervised Jenni & Jin (2021); Dave et al. (2024; 2022a); Thoker et al. (2023); Swetha et al. (2021) and foundational models Bardes et al. (2024a); Wang et al. (2023; 2022); Swetha et al. (2023) capable of handling multiple video understanding tasks, enhancing versatility and generalizability.

**Privacy Preservation in Video Understanding** Recent efforts in video action recognition have addressed visual privacy concerns. Many studies protect visual privacy at the time of data capture by utilizing non-intrusive sensors such as thermal imaging, depth cameras, and event cameras Luo et al. (2018); Hinojosa et al. (2022); Kim et al. (2022); Ahmad et al. (2022; 2023). In this study, we focus exclusively on models using standard RGB cameras. Initial approaches involved reducing the resolution of input data Ryoo et al. (2017); Dai et al. (2015); Liu & Zhang (2020) or employing object detection for targeted obfuscations Ren et al. (2018); Zhang et al. (2021). However, recent research indicates that these methods often fail to balance utility and privacy effectively Wu et al. (2020); Dave et al. (2022b); Fioresi et al. (2023); Kumawat & Nagahara (2022); Peng et al. (2024). Wu et al. (2020) showcased a U-Net Ronneberger et al. (2015)-based adversarial training framework which modifies input frames to decrease the accuracy of private attribute prediction while preserving action recognition performance. Dave et al. (2022b) proposed a self-supervised variant that focuses on reducing mutual information instead of relying on sensitive privacy labels. Fioresi et al. (2023) adapts the self-supervised privacy objective from Dave et al. (2022b) for the anomaly detection task. Compared to the prior input-level anonymization methods our latent-space anonymization method differs in two key aspects: (1) previous methods are tailored to specific downstream tasks, such as action recognition in Dave et al. (2022b) and anomaly detection in Fioresi et al. (2023), while our approach aims to preserve privacy across various downstream video understanding tasks, (2) unlike prior methods, our method does not necessitate the retraining of the video model, thus providing computational efficiency for anonymizing even large-scale video foundation models.

**Bias Mitigation** Computer vision tasks often struggle with spurious correlations Geirhos et al. (2018; 2020), where models rely on irrelevant information to make decisions, such as using background cues for action recognition instead of focusing on subjects' movements Ding et al. (2022); Zou et al. (2023). Unfortunately, biases across a variety of protected demographic attributes, such as perceived gender, skin color, and age Zhao et al. (2017); Stock & Cisse (2017); Buolamwini & Gebru (2018); Wilson et al. (2019); Prabhu & Birhane (2020); Tong & Kagal (2020); Steed & Caliskan (2021); Gustafson et al. (2023); Narnaware et al. (2025); Kim et al. (2025); Swetha et al. (2025); Yousaf et al. (2026); Sirnam et al. (2026) have been found in vision-based tasks. These biases not only skew model performance but can also perpetuate harmful stereotypes. Barbano et al. (2021) explored the relationship between debiasing and privacy preservation, finding that there exists a subset of privacy preservation methods that are suitable for debiasing, giving promise to privacy preservation as a form of debiasing. In contrast to the image domain, biases in the video domain have not been as extensively studied. While a few papers Choi et al. (2019); Li et al. (2023a); Fioresi et al. (2025) address and mitigate scene bias in action recognition tasks, they overlook biases related to human attributes. Motivated by this gap, we introduce, for the first time, protocols to assess gender bias in action recognition. Our findings demonstrate that our self-supervised privacy preservation, even without an explicit bias-related objective, effectively generalizes in mitigating gender bias.

## 3 METHOD

### 3.1 PROBLEM FORMULATION

In this work, we consider handling sensitive issues in video understanding tasks from the dual perspective of privacy preservation and bias mitigation.

**Privacy Preservation** We propose a novel privacy-preserving framework that handles multiple utility tasks across diverse video datasets. Our framework is designed to maintain the high performance of a frozen video encoder across tasks while enforcing robust privacy constraints through the use of an embedding-level anonymization model $f_A$. Specifically, we consider video datasets that span action recognition ($\mathbb{D}_{AR}$), temporal action detection ($\mathbb{D}_{TAD}$), and anomaly detection ($\mathbb{D}_{AD}$).

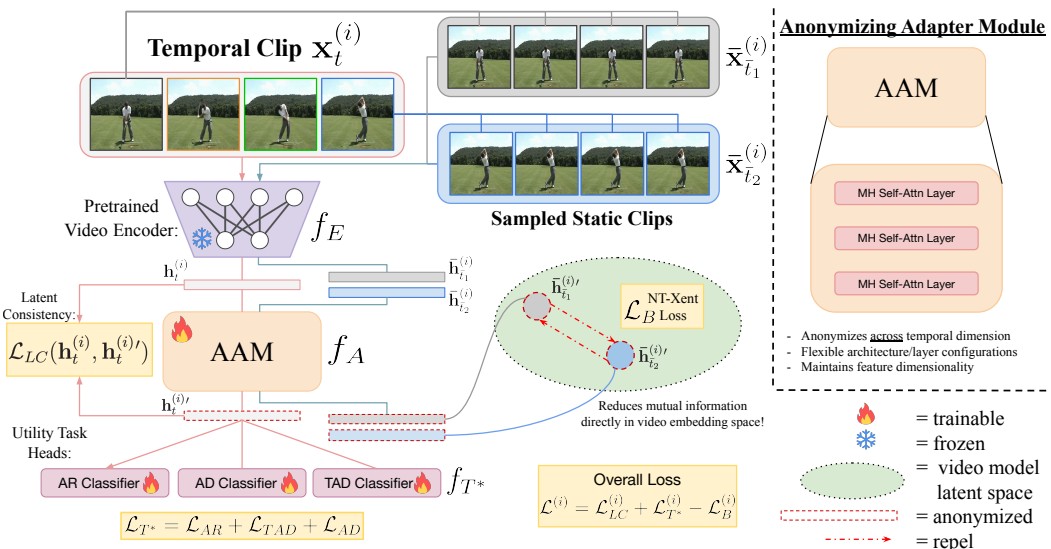

Figure 2: Workflow illustrating the SPLAVU training process. The process begins with a video clip $\mathbf{x}_t^{(i)}$, from which two random frames are sampled to create static clips. All clips are passed through the frozen video encoder $f_E$ to extract global latent video features ([CLS]/average pooled embedding), then further processed by our Anonymization Adapter Module (AAM) $f_A$. The temporal clip features are used for the latent consistency loss and given to the set of task-specific classifier heads $f_{T^*}$. The two static clip features ($\bar{\mathbf{h}}_{t1}^{(i)}$, $\bar{\mathbf{h}}_{t2}^{(i)}$) are utilized in the self-supervised mutual information minimization objective. Gradients from all losses are back-propagated through AAM. A complete training algorithm is provided in Appendix Sec. D.

Each dataset $\mathbb{D}$ contains $N$ video samples, represented as $\{\mathbf{x}^{(i)}, \mathbf{y}^{(i)}\}_{i=1}^N$, where $\mathbf{x}^{(i)}$ is a video instance, and $\mathbf{y}^{(i)}$ is its corresponding task-specific label. We define the set of utility tasks as $\{T_{AR}; T_{TAD}; T_{AD}\} \in T^*$. In implementation, we can choose a subset of tasks from $T^*$ for training, then evaluate on held-out tasks. We introduce a budget privacy evaluation task, denoted as $B$, where performance is measured by private attribute prediction. The framework starts with an off-the-shelf video encoder model $f_E$, left completely frozen. We define $f_E(\mathbf{x}_t^{(i)}) = \mathbf{h}_t^{(i)}$ as the global clip-level embedding, taken as the [CLS] token for transformer encoders or the average-pooled feature for CNN-based encoders. In all experiments, $f_A$ operates only on this single clip-level embedding. The overall goal of $f_A$ is threefold: (1) to maintain the performance of $f_E$ across the set of defined utility tasks $T^*$, (2) to simultaneously reduce the performance on budget private attribute prediction task $B$, and (3) to preserve the general capabilities of $f_E$ such that performance is maintained on *unseen* tasks. This privacy preservation framework is outlined via the following criteria:

*Criterion-1*: Across each utility task, performance should be retained. Specifically, for task $T^n \in T^*$, the loss $\mathcal{L}_{T^n}$ before and after anonymization should be approximately equal.

$$\sum_n^{|T^*|} \mathcal{L}_{T^n}(f_{T^n}(f_A(f_E(X))), Y) \approx \sum_n^{|T^*|} \mathcal{L}_{T^n}(f_{T^n}(f_E(X)), Y). \tag{1}$$

*Criterion-2*: The anonymized encoded features are directly used to compute budget loss $\mathcal{L}_B$ for budget task $B$, which should greatly increase after anonymization.

$$\mathcal{L}_B(f_A(f_E(X))) \gg \mathcal{L}_B(f_E(X)). \tag{2}$$

*Criterion-3*: The anonymization function should maintain the generalization capabilities of $f_E$ by not drastically altering the latent features. Hence, we define a latent consistency objective ($\mathcal{L}_{LC}$) as follows:

$$min\ \mathcal{L}_{LC}(f_A(f_E(X)), f_E(X)). \tag{3}$$

A system that fulfills all of these criterion achieves an effective balance between utility and privacy.

**Perceived Gender Bias Evaluation** In the standard bias evaluation protocol, we are given a video dataset $\mathbb{D}_{AR} = \{(\mathbf{x}^{(i)}, \mathbf{y}^{(i)})\}_{i=1}^{N_{IID}}$, where $\mathbf{x}^{(i)}$ is the $i$th video instance, $\mathbf{y}^{(i)}$ is the associated action label, and $N_{IID}$ is the number of dataset instances. After training, performance is evaluated on unseen bias test set $\mathbb{D}_{reco-OOD} = \{(\mathbf{x}^{(i)}, \mathbf{y}^{(i)})\}_{i=1}^{N_{OOD}}$, where $N_{OOD}$ is the number of out-of-distribution instances. The aim of any debiasing technique is to learn generalizable features of $\mathbb{D}_{IID}$ such that performance is maximized on $\mathbb{D}_{OOD}$ without compromising IID performance.

When considering perceived gender information, our in-distribution video dataset is now formulated as $\mathbb{D}_{AR} = \{(\mathbf{x}^{(i)}, \mathbf{y}^{(i)}, \mathbf{g}^{(i)})\}_{i=1}^{N_{IID}}$, where $\mathbf{g}^{(i)} \in \{male, female\}$ is the perceived binary gender label. We acknowledge that this binary formulation is not ideal and not inclusive of all gender categories. The bias evaluation test set also includes label $\mathbf{g}$ in order to evaluate subclass performance. Final evaluation is on a test set with a different label distribution.

## 3.2 ANONYMIZATION FRAMEWORK

This section describes the full anonymization framework and training. The framework consists of 3 major components: (1) a frozen video encoder backbone $f_E$, (2) an anonymization function adapter $f_A$, which modifies the latent features while retaining the original shape, and (3) a set of utility classifier heads $\{f_{T_{AR}}; f_{T_{TAD}}; f_{T_{AD}}\} \in f_{T^*}$ for a predefined set of tasks. Note that $f_A$ acts solely on the global clip embedding $\mathbf{h}_t$ used as input to each utility task head.

**Network Initialization** To start, we initialize $f_A$ to act as an identity function. The video encoder model $f_E$ is initialized with off-the-shelf weights of Kinetics400 Carreira & Zisserman (2017) pre-training. Each $f_T$ classifier head matches a standard architecture for the provided task. For stability, these are initialized through non-anonymized training on their respective utility tasks. For action recognition, $f_{T_{AR}}$ is a simple linear layer. For action detection $f_{T_{TAD}}$ and anomaly detection $f_{T_{AD}}$ architectures, TriDet Shi et al. (2023) and MGFN Chen et al. (2023) respectively are utilized.

## Anonymization Training

The training process consists of an adversarial optimization between a budget privacy loss $\mathcal{L}_B$ and a collection of standard utility losses $\mathcal{L}_{T^*}$, regularized by a proposed latent consistency loss $\mathcal{L}_{LC}$.

**Collaborative Utility Objectives** To retain the action understanding capabilities of the pretrained model, we employ a co-training framework where multiple tasks collaborate to optimize performance. The action classifier head, $f_{T_{AR}}$, is trained using the standard cross-entropy loss. Our latent formulation enables, for the first time, anonymization training using gradients from alternate downstream utility tasks. As such, we integrate training objectives from state-of-the-art approaches in TAD and AD. More detailed information can be found in Appendix Sec. B. The utility losses from these tasks are combined and jointly optimized through the following:

$$\mathcal{L}_{T^*}^{(i)} = \omega_{AR}\mathcal{L}_{AR} + \omega_{TAD}\mathcal{L}_{TAD} + \omega_{AD}\mathcal{L}_{AD}, \tag{4}$$

where $\omega$ represents a hyperparameter controlling the relative weight of each task's loss objective. We set $\omega_{AR} = \omega_{TAD} = \omega_{AD} = 1$ to balance task contribution. Notably, we ablate the set of tasks chosen for training (ex: $\omega_{AD} = 0$, Tab. 5), finding that SPLAVU indeed generalizes to unseen tasks when using our latent consistency loss.

**Clip-Level Budget Privacy Objective** Our clip-level self-supervised budget privacy objective is the key component for facilitating anonymization without requiring private attribute labels. The intuition is that two frames share a lot of mutual information, so if we *minimize* the similarity between them, the shared spatial information gets destroyed. A crucial difference setting SPLAVU apart from prior works is that the anonymizer works across the temporal dimension using 3D clip features instead of a 2D U-Net Wu et al. (2020); Dave et al. (2022b); Fioresi et al. (2023). This way, when combined with utility task losses, the anonymization model learns to remove all spatial information, maintaining only temporal information useful for solving the utility task. We utilize the SimCLR NT-Xent Chen et al. (2020) contrastive loss as $\mathcal{L}_B$, defined as follows:

$$\mathcal{L}_B^{(i)} = -log \frac{d(\bar{\mathbf{h}}_{t_1}^{(i)}, \bar{\mathbf{h}}_{t_2}^{(i)})}{\sum_{j=1}^{N}[\mathbb{1}_{[j \neq i]}d(\bar{\mathbf{h}}_{t_1}^{(i)}, \bar{\mathbf{h}}_{t_1}^{(j)}) + d(\bar{\mathbf{h}}_{t_1}^{(i)}, \bar{\mathbf{h}}_{t_2}^{(j)})]}, \tag{5}$$

where $\bar{\mathbf{h}}_t^{(i)}$ represents the feature vector of a static clip sampled from video $\mathbf{x}^{(i)}$ at time $t$, $d(u, v) = exp(u^T v/(\|u\|\|v\|\tau))$ computes the similarity between the input vectors with temperature parameter $\tau$. $\mathbb{1}_{[j \neq i]}$ is an indicator function that equals 1 when $j \neq i$. Minimizing this loss increases the similarity between inputs $\bar{\mathbf{h}}_{t_1}^{(i)}$ and $\bar{\mathbf{h}}_{t_2}^{(i)}$ over the sum of all other clips in the batch. Instead, we opt to *maximize* the loss, resulting in the objective destroying mutual information between these clips instead. Notably, as opposed to previous works Wu et al. (2020); Dave et al. (2022b); Fioresi et al. (2023), we do not use a disjoint image encoder model to compute $\bar{\mathbf{h}}_t^{(i)}$. Alternatively, the video encoder model $f_E$ itself is used to process clip frames (tiled to match a standard clip shape, see Figure 2), leading to a much more natural utility-privacy interaction for improved anonymization. These static clip features are then directly utilized in this budget privacy loss (Eq. (5)).

**Latent Consistency Objective** Early experiments with the privacy and utility losses indicated that the anonymization process tends to overfit to the proxy-utility tasks used in training (see Tab. 6), compromising its effectiveness on unseen tasks. Consequently, the primary motivation behind introducing our latent consistency objective is to ensure that the anonymization learned by the model remains generalizable and is not biased toward the specific utility task(s) it is trained on. This can be accomplished by regularizing the anonymization to preserve the general latent structure of the utility encoder $f_E$. To this end, we propose a latent consistency loss that encourages the model to preserve important latent features while still achieving privacy preservation:

$$\mathcal{L}_{LC}^{(i)} = \|f_E(\mathbf{x}^{(i)}) - f_A(f_E(\mathbf{x}^{(i)}))\|_2^2, \tag{6}$$

where $\mathbf{x}^{(i)}$ is the input video clip and $\| \cdot \|_2^2$ is the $\ell_2$ distance. This key component ensures that the anonymization does not shift $f_E$ features completely into a new space that is overfit to the utility training tasks, leading to well-generalizing anonymization.

**Overall Training Objective** $f_A$ and $f_T$ are jointly optimized utilizing the following compound loss:

$$\mathcal{L}^{(i)} = \omega_{LC} * \mathcal{L}_{LC}^{(i)} + \omega_T * \mathcal{L}_{T*}^{(i)} - \omega_B * \mathcal{L}_B^{(i)}, \tag{7}$$

where $\omega_{LC}$, $\omega_T$, and $\omega_B$ are weights to control the strength of each objective. The privacy term is maximized via the negative sign in $-\omega_B * \mathcal{L}_B$ and works against $\mathcal{L}_T$ and $\mathcal{L}_{LC}$ in a GAN-style paradigm, until the anonymizer is able to remove all encoded spatial information except for what is necessary for performance on the utility tasks. After this training, we are left with a lightweight anonymization adapter $f_A$ that can be appended to the off-the-shelf video encoder model $f_E$ for use in a variety of downstream tasks.

**Anonymizing Adapter Module (AAM)** To carry out latent anonymization, we propose the use of an anonymizing adapter module. AAM is applied to *clip-level* features, allowing for reasoning across the temporal dimension, better aligning with utility tasks. Given latent feature $\mathbf{h}^{(i)} = f_E(\mathbf{x}^{(i)})$, AAM is trained to modify $\mathbf{h}^{(i)}$ with the above loss objective. We utilize a multi-head self-attention-based transformer encoder for AAM. A design choice ablation can be found in Appendix Sec. C.

## 4 EVALUATION PROTOCOLS

To ensure that our anonymization preserves the utility of the off-the-shelf encoder across multiple tasks, we evaluate its performance comprehensively. Existing anonymization methods, which typically use action recognition as the sole proxy utility task, significantly degrade performance on alternate downstream tasks. However, pretrained models are known to demonstrate strong performance in areas like temporal action detection and anomaly detection. Therefore, we assess the learned features across *five distinct tasks* to thoroughly evaluate their effectiveness post-anonymization.

### 4.1 PRIVACY EVALUATION

First, we employ an established privacy preservation protocol to ensure that $f_A$ removes sensitive-attribute related information. Although we focus on action-related video understanding models, private attribute information is still exposed in the latent features of the backbone encoder. The VISPR dataset Orekondy et al. (2017) evaluates privacy-preservation by measuring performance on a multi-class classification problem for various sensitive visual private attributes, with performance measured by mean average precision across classes (cMAP). For evaluation, we train a classifier

on static clip representations, adhering to protocols from prior work in video privacy Dave et al. (2022b). Additionally, we evaluate on *temporal-based* VP-HMDB51 and VP-UCF101 datasets Li et al. (2023b), which contain both action and private attribute labels. Here, the full clip representations are used for attribute classification with cMAP as the evaluation metric. Since the goal is to enhance privacy, not to accurately predict attributes, a lower performance indicates better privacy preservation. For further temporal analysis, we evaluate the effect of anonymization on a *motion-based* sensitive attribute, namely gait recognition on Casia-B Yu et al. (2006).

## 4.2 UTILITY VIDEO TASK EVALUATION

**Action Recognition** Action recognition involves analyzing spatio-temporal video features to classify actions. In our framework, features are extracted from all videos using a Kinetics-pretrained video encoder. Then, a linear classifier is trained for evaluation on each dataset $\mathbb{D}_{AR}$, namely Kinetics400 Carreira & Zisserman (2017), UCF101 Soomro et al. (2012), and HMDB51 Kuehne et al. (2011). Additional action results are shown on VP-HMDB51 and VP-UCF101 Li et al. (2023b). Evaluation is top-1 accuracy on 5 evenly spaced clips from each test video.

**Temporal Action Detection** Temporal action detection (TAD) involves identifying the specific time intervals within an untrimmed video where particular actions occur. TAD utilizes features from a Kinetics-pretrained video encoder model. Given $\mathbb{D}_{TAD}$, $f_A$ is used to generate anonymized feature set $\mathbb{F}_{TAD} = \{ f_A(f_E(X^{(i)})) \mid \forall X^{(i)} \in \mathbb{D}_{TAD} \}$. Our TAD evaluation uses THUMOS14 Jiang et al. (2014) as $\mathbb{D}_{TAD}$. We choose one of the recent state-of-the-art methods, TriDet Shi et al. (2023) with default hyperparameters to evaluate using mean Average Precision (mAP).

**Weakly-Supervised Anomaly Detection** Weakly supervised anomaly detection (WSAD) involves localizing timestamps of anomalous (unexpected) events given long, untrimmed videos and only video-level labels. Our evaluation uses UCF-Crime Sultani et al. (2018) as $\mathbb{D}_{AD}$. Given $\mathbb{D}_{AD}$, $f_A$ is used to create an anonymized feature set $\mathbb{F}_{AD} = \{ f_A(f_E(X^{(i)})) \mid \forall X^{(i)} \in \mathbb{D}_{AD} \}$. A recent state-of-the-art anomaly detection method MGFN Chen et al. (2023) is used with default hyperparameters. Final evaluation is given as a frame-level ROC AUC percentage.

## 4.3 GENDER PRESENTATION BIAS PROTOCOLS

**NTU Bias Evaluation** We further verify anonymization efficacy by evaluating on our proposed attribute bias protocols. The NTU60 Shahroudy et al. (2016) action recognition dataset is curated to have minimal scene and subject biases as each actor performs each action in different scenes. Given that each video is labeled with subject ID, we can introduce an artificial bias by controlling the gendered subclass ratios across actions. A ratio of 95% Sagawa et al. (2019) is set for all but one action, where the typical ratio is inverted to create a spurious shortcut for the model. This is done for each gender, resulting in two subsets: NTU-Bias-F and NTU-Bias-M. Detailed protocol creation info is found in Appendix Sec. A.

**Toyota Smarthome Bias Evaluation** Unlike NTU, the Toyota Smarthome (TSH) dataset is naturally imbalanced and represents a real-world scenario with elderly individuals performing daily activities. Each video is labeled with a subject ID, allowing for robust evaluation of perceived gender biases without using a gender classifier. Here, we look at the performance of each gender subclass. A model is considered less biased if the baseline gap between the subclass accuracies is reduced.

## 5 EXPERIMENTS

Further dataset and implementation details can be found in Appendix Sec. A and B , respectively.

## 5.1 MAIN EVALUATION: PRIVACY VS TASK TRADEOFFS

Our evaluation of the proposed method covers private attribute prediction protocols and a variety of downstream tasks. We observe in Table 1 that our approach consistently generalizes well across all tasks, closely maintaining the performance of the non-anonymized videos. In contrast, previous methods struggle to preserve performance uniformly across tasks, evident in the temporal action detection results of Wu et al. (2020); Dave et al. (2022b); Fioresi et al. (2023). Table 2 demonstrates

Table 1: Performance of anonymization methods across a downstream task evaluation suite. Method in gray trains using private attribute labels. Our method achieves a strong improvement in privacy-preservation with minimal reduction in task performance.

| Anonymization Method | Network | Privacy | Action Recognition | Action Recognition | Temporal Action Detection | Anomaly Detection |
|---|---|---|---|---|---|---|
| | | VISPR cMAP (↓) | Kin400 Top-1 (↑) | UCF101 Top-1 (↑) | THUMOS14 mAP(↑) | UCF Crime AUC (↑) |
| Raw Videos | | 63.64 | 62.67 | 90.30 | 25.29 | 77.68 |
| Downsample-2x | | 55.64 | – | 81.78 | 16.94 | 76.09 |
| Downsample-4x | | 52.84 | – | 66.21 | 15.72 | 68.12 |
| Blurring | | 58.68 | – | 83.90 | 17.65 | 75.69 |
| Blackening | I3D | 56.36 | – | 68.62 | 15.72 | 73.91 |
| VITA TPAMI'20 | | 54.72 | – | 75.83 | 16.10 | 73.74 |
| SPAct CVPR'22 | | 55.60 | 46.93 | 75.77 | 16.20 | 73.93 |
| TeD-SPAD ICCV'23 | | 52.30 | 47.20 | 76.64 | 17.27 | 74.81 |
| **Ours** | | 41.07↓35.5% | 62.11↓0.9% | 90.14↓0.2% | 24.92↓1.5% | 75.69↓2.6% |
| Raw Videos | VideoMAE-B | 70.47 | 74.86 | 96.80 | 60.82 | 85.79 |
| **Ours** | | 49.92↓28.9% | 74.23↓0.8% | 96.11↓0.7% | 60.50↓0.5% | 85.08↓0.8% |
| Raw Videos | VJEPA-H | 72.44 | 77.03 | 97.67 | 66.66 | 85.79 |
| **Ours** | | 51.42↓29.0% | 76.62↓0.5% | 97.54↓0.1% | 66.30↓0.4% | 84.81↓1.1% |

Table 2: Accuracy and privacy evaluation using video-level private attribute protocols VP-HMDB51 and VP-UCF101 Li et al. (2023b), evaluated as top-1 accuracy and cMAP. Our anonymizer successfully mitigates temporal private attribute prediction, matching the quality of a latent anonymizer trained with a supervised privacy loss Wu et al. (2020).

| Anonymization Method | VP-HMDB51 | | VP-UCF101 | |
|---|---|---|---|---|
| | Top-1 Acc.↑ | cMAP↓ | Top-1 Acc.↑ | cMAP↓ |
| Raw Videos | 72.6 | 76.4 | 96.8 | 75.9 |
| Supervised | 72.4 | 70.4 | 96.5 | 69.5 |
| **Ours** | 72.1 | 70.5 | 96.8 | 69.6 |

resistance to temporal private attributes and that performance matches that of a version trained with supervised attribute labels. Experiments with large VFMs see similar performance trends, confirming the efficacy and scalability of SPLAVU. Results in Table 2 demonstrate the ability of our method to handle temporal private attribute settings.

## 5.2 GENDER BIAS EVALUATION

The first row of Table 3 shows the performance difference between each gender presentation subclass in the NTU-Bias-F protocol, where the action *brush_hair* is chosen as the gendered shortcut action label. The baseline performance disparity between perceived gender subclasses is an unacceptably large 9.42%. Applying latent anonymization impressively reduces this gap by a relative **42.3%**. The second row includes results for the complementary protocol NTU-Bias-M (also *brush_hair* shortcut). Interestingly, the baseline subclass performance disparity is less than that of NTU-Bias-F (5.00%), but our method is still capable of reducing this unfair split and improving overall performance.

To confirm that these observations hold true in a real-world setting, we look at the final row of Table 3 to see the performance on the TSH Das et al. (2019) protocol. Notably, our method improves the both the classifier quality and fairness. In this realistic scenario with a naturally occurring bias, SPLAVU reduces the gap between perceived gender subclasses by an astonishing relative **39.5%**.

## 5.3 ABLATIONS AND ANALYSIS

We utilize the VideoMAE-B model for all ablations. Further details can be found in Appendix Sec. C.

**Effect of task-specific training:** Our important ablation in Table 5 demonstrates the effects of training our anonymizer without specific tasks. Notably, the highlighted cells show impressive gen-

Table 3: Bias evaluation across gendered groups; anonymization reduces subclass accuracy gaps.

| Dataset | Method | P. Female Acc. (%) | P. Male Acc. (%) | Overall Acc. (%) | Δ Subclass Acc. (%) |
|---|---|---|---|---|---|
| NTU-Bias-F | Baseline | 46.78 | **56.20** | 51.49 | 9.42 |
| | Ours | **49.91** | 55.35 | **52.63** | **5.44** |
| NTU-Bias-M | Baseline | **55.23** | 50.23 | 52.78 | 5.00 |
| | Ours | 55.07 | **51.04** | **53.06** | **4.03** |
| TSH | Baseline | 65.15 | **70.90** | 67.02 | 5.75 |
| | Ours | **66.51** | 69.99 | **67.64** | **3.48** |

Table 4: Performance comparison across video understanding protocols using anonymization models pretrained on datasets of varying scale (leftmost column). $f_A$ is trained using action recognition as the only utility task. Each row shows the anonymization pretraining dataset, while columns show downstream evaluation tasks. Learning an anonymizer on small datasets such as HMDB51 maintains an impressive privacy-utility tradeoff across tasks compared to raw, non-anonymized data.

| Pretraining Dataset | VISPR cMAP (↓) | K400 Top-1 (↑) | UCF101 Top-1 (↑) | HMDB51 Top-1 (↑) | ToyotaSH Top-1 (↑) | UCF-Crime AUC (↑) | THUM14 mAP (↑) |
|---|---|---|---|---|---|---|---|
| Raw Data | 70.47 | 74.86 | 96.80 | 72.94 | 65.05 | 85.79 | 60.82 |
| K400 | 52.57 | **74.74** | 96.11 | 71.51 | 65.34 | 83.47 | 56.45 |
| UCF101 | **49.64** | 74.49 | **97.01** | 72.68 | 62.29 | 84.14 | 52.18 |
| HMDB51 | 54.35 | 74.55 | 96.56 | **73.92** | 65.82 | **84.52** | **56.50** |
| Toyota SH | 51.58 | 74.35 | 96.09 | 72.42 | **67.27** | 74.92 | 41.30 |

eralization to unseen tasks with just a minor drop in performance compared to training on them. For example, looking at row (c) shows $f_A$ training with only action detection, yet the performance on action recognition and anomaly detection remain within **1.3%** of the non-anonymized score. Across the board, thanks to the latent consistency loss, performance is not dependent on having seen the given utility task during training, proving that SPLAVU can effectively *generalize to unseen tasks*. See Appendix Table 13 for additional generalization results on the video retrieval task.

Table 5: Ablation on tasks seen during anonymization training. The checkmark (✓) labels seen tasks, x-mark (✗) and highlighted cells indicate tasks unseen during training. Performance generalizes to unseen tasks, while directly training further improves results.

| Training Tasks | | | Evaluation Tasks | | | |
|---|---|---|---|---|---|---|
| AR | TAD | AD | VISPR cMAP (↓) | K400 Acc. (↑) | THUM14 mAP (%) (↑) | UCF-Crime AUC (%) (↑) |
| (a) ✗ | ✗ | ✗ | 70.47 | 74.86 | 60.82 | 85.79 |
| (b) ✓ | ✗ | ✗ | 52.57 | 74.65 | 56.45 | 83.47 |
| (c) ✗ | ✓ | ✗ | 50.17 | 73.86 | 58.80 | 83.67 |
| (d) ✗ | ✗ | ✓ | 49.34 | 73.51 | 57.34 | 84.56 |
| (e) ✓ | ✓ | ✗ | 48.74 | **74.30** | 58.67 | 83.88 |
| (f) ✓ | ✗ | ✓ | 50.77 | 74.24 | 58.83 | 84.28 |
| (g) ✗ | ✓ | ✓ | **48.01** | 73.70 | 60.41 | 84.77 |
| (h) ✓ | ✓ | ✓ | 49.92 | 74.23 | **60.50** | 85.08 |

Table 6: Ablation on training losses. Without latent consistency, the anonymizer overfits to the action recognition task.

| $\mathcal{L}_T$ | $\mathcal{L}_B$ | $\mathcal{L}_{LC}$ | VISPR cMAP (↓) | HMDB51 Top-1 Acc. (↑) | THUM14 mAP (%) (↑) |
|---|---|---|---|---|---|
| ✗ | ✗ | ✗ | 70.47 | 74.20 | 60.82 |
| ✗ | ✓ | ✓ | 45.12 | 4.71 | 1.52 |
| ✓ | ✗ | ✓ | 70.44 | 73.17 | 60.34 |
| ✓ | ✓ | ✗ | 51.70 | 72.88 | 3.81 |
| ✓ | ✓ | ✓ | **54.35** | **73.92** | **56.50** |

Table 7: Gait recognition experiment on Casia-B. Latent consistency (LC) controls recognition of *temporal* private attributes.

| Method | VISPR | Casia-B |
|---|---|---|
| Baseline | 70.47 | 69.73 |
| Ours (w/ LC) | 54.35 | 53.45 |
| Ours (w/o LC) | 51.70 | 26.67 |

**Effect of training set scale:** To evaluate the scaling of our anonymization method, we perform all downstream tasks while varying the size of training datasets as shown in Table 4. We see that SPLAVU demonstrates impressive data-efficiency by generalizing to all downstream tasks, even when training on *small-scale* datasets like HMDB51.

**Temporal sensitive attribute recognition:** Table 7 shows the performance of our model on the retrieval-based gait recognition task with no training. Because gait recognition benefits from understanding a temporal signature, latent consistency does *not* suppress potentially sensitive temporal

Table 8: Feature inversion attack results. Our model demonstrates strong robustness to black-box attacks, where action performance is recovered but not private attribute prediction. Random chance on each dataset is included to contextualize performance.

| Method/Attack | VP-HMDB51 | | VP-UCF101 | |
|---|---|---|---|---|
| | Top-1 Acc.↑ | cMAP↓ | Top-1 Acc.↑ | cMAP↓ |
| Raw Videos | 72.6 | 76.4 | 96.8 | 75.9 |
| Random Chance | 2.0 | 58.3 | 1.0 | 58.7 |
| **Ours** | 72.1 | 70.5 | 96.8 | 69.6 |
| Black-Box Inversion | 67.3 | 58.8 | 96.8 | 58.4 |
| White-Box Inversion | 68.0 | 73.7 | 94.5 | 72.6 |

attributes. If task overfitting is not a concern, not using latent consistency properly defends against private temporal attribute recognition. Otherwise, a temporal sensitive attribute prediction task can be additionally included in the budget privacy loss Wu et al. (2020) to maintain generalization.

**Framework loss component ablation:** Our ablation study examines key training losses of the anonymization process in Table 6. Action recognition is the only training utility task to evaluate task generalization. Unsurprisingly, omitting the utility loss leads to a considerable drop in model performance. Excluding the privacy budget objective results in no privacy gains over the baseline, emphasizing its necessity. Furthermore, removing latent consistency loss affects unseen task performance, whereas seen task (action recognition) performance is maintained. This underscores the importance of the latent consistency loss in ensuring generalization of our anonymization method.

**Attack resistance:** A major concern of any anonymization framework is the potential for an inversion attack. Table 8 includes results from white and black-box attack settings on VP-HMDB51 and VP-UCF101 Li et al. (2023b). In the *black-box* setting, the attacker observes input-output pairs of the anonymizer and trains a separate inversion model where no gradients through $f_A$ are available. In the *white-box* setting, the attacker has full access to $f_A$ and instead performs gradient-based reconstruction by optimizing a synthetic embedding to match the observed anonymized output. In both cases, recovered embeddings are evaluated by feeding them into downstream models trained on non-anonymized features to measure action accuracy and privacy cMAP. Notably, in the black-box setting, the inversion network successfully recovers action classification performance, yet does not improve attribute classification performance above random chance. The model is not robust to white box attacks, which are typically unrealistic in deployment scenarios. Robustness to white-box attacks is a promising avenue for future work. See Appendix Sec. C and Figure 4 for qualitative analysis on reconstruction-based attacks, where our anonymization prevents faithful video reconstruction.

**Limitations:** The latent anonymization framework cannot mitigate a threat that may occur if videos must be transmitted before feature extraction. We leave such cases to complementary techniques like secure transmission, or to prior anonymization works if optimal accuracy is not a concern. Our formulation currently targets global-embedding tasks (e.g., action recognition, temporal action detection, anomaly detection, retrieval). Extending anonymization to dense tasks such as segmentation or captioning that require patch-wise embeddings is an important direction for future work.

## 6 CONCLUSION

We propose an innovative privacy-preserving method via a novel formulation of latent space anonymization called SPLAVU. Our method is the first to enable generalized anonymization for unprecedented performance across various downstream video understanding tasks, including action recognition, anomaly detection, and temporal action detection. It employs a clip-level self-supervised privacy budget within the latent space, coupled with a latent consistency loss to maintain its powerful generalization capability. Furthermore, our novel protocols for assessing gender bias contribute to the development of more responsible and unbiased video understanding models.

## 7 Acknowledgement

This work was supported in part by the National Science Foundation (NSF) and Center for Smart Streetscapes (CS3) under NSF Cooperative Agreement No. EEC-2133516.

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

SUPPLEMENTARY OVERVIEW

# A  DATASET DETAILS

**Kinetics400 Carreira & Zisserman (2017)** is a large-scale video action dataset of YouTube videos which includes 400 human action classes with at least 400 video clips for each action. Each clip lasts around 10 seconds and is labeled with a single action class. The dataset is widely used for pretraining deep learning models for use in many video understanding tasks.

**UCF101 Soomro et al. (2012)** is an action recognition dataset of realistic action videos consisting of 101 action categories. With over 13,000 videos from various actions and scenes, it provides a diverse set of actions and a broad range of variability in terms of actions, viewpoints, appearances, and backgrounds.

**HMDB51 Kuehne et al. (2011)** is a collection of 6,766 video clips distributed across 51 human action categories, each containing a minimum of 101 clips. The dataset includes a wide range of human actions and is designed for the development and evaluation of action recognition methods.

**NTU RGB+D 60 Shahroudy et al. (2016)** is a large-scale multi view human action recognition dataset complete with RGB video, depth maps, and skeleton joints, and IR sequences. This work only uses the RGB frames. Each of the 40 subjects are recorded completing 60 daily activities from 3 different cameras.

**Toyota Smarthome Das et al. (2019)** is a challenging real-world activity classification dataset captured from 7 independent Kinect v1 cameras. The clips recorded 18 senior subjects performing 31 daily activities in a natural manner. This work only uses the provided RGB frames. The dataset contains high class imbalance, intra-class variation, and duration variance.

**THUMOS14 Jiang et al. (2014)** focuses on temporal action localization in untrimmed videos. It extends the UCF101 dataset with temporal annotations for a subset of the action classes, providing detailed temporal annotations for 20 action classes across 200 validation videos and 213 test videos.

**UCF-Crime Sultani et al. (2018)** is a large-scale dataset of surveillance videos designed for anomaly detection. It consists of 1,900 long and untrimmed videos for a total of 128 hours. The videos contain examples of 13 different real-world anomalies, including burglary, robbery, and fighting, among others, making it suitable for training and evaluating video anomaly detection models.

**VISPR Orekondy et al. (2017)** consists of around 22,000 Flickr images annotated with 68 privacy-related attributes such as gender, age group, skin color, and more. It offers a multi-class classification protocol for assessing private attribute prediction. Table 9 shows the VISPR attribute split used, which we have adopted from Wu et al. (2020); Dave et al. (2022b); Fioresi et al. (2023).

**Casia-B Yu et al. (2006)** is a gait recognition dataset consisting of 13,640 video clips of 124 subjects. We utilize it to evaluate the effect of anonymization on a *temporal-based* sensitive attribute, namely gait. Evaluation is through top-1 retrieval of a gallery video with the same subject. The full dataset is used for evaluation, with the first 4 repetitions of each walk in the gallery and the last 2 per subject used as the probe.

**VP-HMDB51/VP-UCF101 Li et al. (2023b)** are versions of the base HMDB51 Kuehne et al. (2011) and UCF101 Soomro et al. (2012) datasets annotated with five clip-based visual private attribute labels, following the VISPR label setting.

**Proposed NTU Bias Evaluation Details** More formal details for the creation of the proposed perceived gender NTU bias protocol are described here. While the original dataset is balanced in terms of scene and actor, the distribution of actor/video counts are not balanced with respect to perceived gender. To properly evaluate bias mitigation, it is essential to ensure that there are no performance

Table 9: Privacy attributes from subset of VISPR Orekondy et al. (2017) labels as used in previous works.

| VISPR1 Wu et al. (2020); Dave et al. (2022b); Fioresi et al. (2023) | |
| --- | --- |
| **Label** | **Description** |
| a17_color | skin color |
| a4_gender | gender |
| a9_face_complete | full face visible |
| a10_face_partial | part of face visible |
| a12_semi_nudity | partial nudity |
| a64_rel_personal | shows personal relationship |
| a65_rel_soci | shows social relationship |

differences stemming from the larger number of male subjects and training videos. The subject IDs are used to first restructure the dataset in an effort to maximize fairness across the gender subgroups. As such, within themselves, the train and test sets should contain both an even number of male and female subjects AND an even number of videos per action. Formally, let's take the set of subjects $S = \{s_i\}_{i=1}^{N_S}$, where $N_S$ is the number of subjects in the dataset. For each subject $s_i \in S$, there is an associated gender label $\mathbf{g}(s_i)$ where $\mathbf{g}(s_i) \in \{male, female\}$. We set $N_m = N_f = \frac{N_S}{2}$, where $N_m$ and $N_f$ are the number of male and female subjects, respectively. Using the above notation with $\mathbb{D}_{IID}$ abbreviated to $D$, we define $D_m = \{(\mathbf{x}_i, \mathbf{y}_i, \mathbf{g}_i) \in D | \mathbf{g}_i = male\}$ and $D_f = \{(\mathbf{x}_i, \mathbf{y}_i, \mathbf{g}_i) \in D | \mathbf{g}_i = female\}$. We set $|D_m| = |D_f| = \frac{|D|}{2}$. With the dataset balanced across subject counts, subject genders, video count per action/gender, and background representation, the model should not have access to simple bias shortcuts.

To directly measure gender presentation bias, we inject an artificial bias related to perceived gender by creating a simple spurious shortcut for the model to follow. Specifically, we control the subclass ratios across all actions, setting $P(\mathbf{g}(s) = male|\mathbf{y}) = 0.95$ and $P(\mathbf{g}(s) = female|\mathbf{y}) = 0.05$, following the correlation strength in Sagawa et al. (2019). However, for one action chosen at random, we flip this ratio, keeping 95% of perceived female videos ($P(\mathbf{g}(s) = female|\mathbf{y}) = 0.95$) and only 5% of perceived male videos ($P(\mathbf{g}(s) = male|\mathbf{y}) = 0.05$). We refer to this subset as NTU-Bias-F. To ensure that the shortcut taking is gender presentation agnostic, we repeat this protocol by swapping the subclasses, creating NTU-Bias-M. We find that swapping this subclass ratio for one action class reduces overall performance and causes a gap in subclass performance.

## B    IMPLEMENTATION DETAILS

All of our code is implemented in PyTorch Paszke et al. (2019). In this section, we clarify notation and provide implementation details regarding network architecture, input preprocessing, hyperparameters, and training schedules.

### B.1    CLARIFYING GLOBAL EMBEDDING NOTATION

In the main paper, we use the notation $\mathbf{h}_t = f_E(\mathbf{x}_t)$ to denote the *global* clip-level embedding produced by the frozen video encoder $f_E$. In standard practice, transformer-based video encoders (e.g., VideoMAE, ViViT) produce a sequence of tokens prepended by a special [CLS] token: $f_E(\mathbf{x}_t) = [\mathbf{h}_t^{CLS}; \mathbf{h}_t^0; \ldots; \mathbf{h}_t^N]$, where $\mathbf{h}_t^{CLS}$ is the global classification token and $\mathbf{h}_t^0, \ldots, \mathbf{h}_t^N$ denote patchwise spatio-temporal tokens. In CNN-based video encoders (e.g., I3D, R3D), the encoder processes the input clip through a hierarchy of spatiotemporal convolutional blocks, producing a feature tensor $f_E(\mathbf{x}_t) = \mathbf{F}_t \in \mathbb{R}^{T' \times H' \times W' \times C}$, which is then aggregated into a global clip-level representation via spatiotemporal average pooling: $\mathbf{h}_t^{\text{pool}} = \text{AvgPool}(\mathbf{F}_t) \in \mathbb{R}^C$. This pooled feature $\mathbf{h}_t^{\text{pool}}$ serves as the analog to the [CLS] token in transformer-based encoders. Because our anonymizer operates on the unified notion of a single clip-level representation, either the [CLS] token or the globally pooled CNN feature, we simplify notation by writing $f_E(\mathbf{x}_t)$ to refer to this global embedding regardless of encoder type.

## B.2 Network Architecture

Each video encoder $f_E$ model is left unchanged from the original implementation. The $f_{T_{AR}}$ classifier head is a simple linear layer `Linear(d, N)`, where $d$ is the feature vector dimension of $f_E$ and $N$ is the number of classes in $\mathbb{D}_{AR}$. In the temporal action detection and anomaly detection task, classifier implementations are unmodified from the original TriDet Shi et al. (2023) and MGFN Chen et al. (2023) works, respectively. For the private attribute prediction task, a 2-layer MLP is used: `Linear(d, d) → Linear(d, 7)` with a ReLU activation after the first layer. For the $f_A$ AAM, we ablate different architecture styles (see Table 10). To break it down, we tried standard MLPs of different depths and self-attention based adapters of different depths. Each MLP layer is composed of a `Linear(d, d)` followed by a ReLU activation and a BatchNorm1D layer, and dropout with a probability of 0.1 during training. The self-attention layers are standard Multi-headAttention blocks with dim $d$ and 8 heads by default, with the head count ablated in Table 12.

## B.3 Inputs and Augmentations

All inputs consist of 16 frame clips sampled with consecutive frames, resized to spatial resolution of $224 \times 224$. For training, only random resized crop and random horizontal flip with probability 50% are utilized. In validation, the short edge is resized to 256, then a center crop of $224 \times 224$ is taken. Standard ImageNet Krizhevsky et al. (2012) mean and standard deviation based normalization is applied in both settings. The input and augmentation protocol is consistent for every $f_E$.

## B.4 Training Details and Hyperparameters

Each AAM variation is trained using an $\ell_1$ loss to reconstruct the input features for 100 epochs with the AdamW Loshchilov & Hutter (2017) optimizer and a learning rate of 2e-5. Kinetics400 Carreira & Zisserman (2017) features are used as the train-test set. Privacy evaluation is carried out using supervised training of the predictor MLP for 100 epochs at a learning rate of 1e-3. A learning rate scheduler is based on the loss plateau where it decreases the learning rate to 1/5th.

For anonymization training, the base learning is 1e-4 for both $f_A$ and $f_{T^*}$, corresponding to a batch size of 512, scaled when necessary according to the linear scaling rule Goyal et al. (2017a). By default, $\omega_{LC} = 100$, $\omega_T = 1$, and $\omega_B = 1$ (Main Equation (7)). The anonymization training is carried out for 100 epochs.

## B.5 Additional Collaborative Task Losses

Here we further define the integrated task losses $\mathcal{L}_{TAD}$ and $\mathcal{L}_{AD}$ referenced in Main Paper Section 3.2.

**Temporal Action Detection Loss $\mathcal{L}_{TAD}$ (TriDet Shi et al. (2023)):**

The overall TriDet loss function combines classification and regression components and is defined as:

$$\mathcal{L}_{TAD} = \frac{1}{N_{pos}} \sum_{l,t} \mathbb{1}_{\{c_t^l > 0\}} \left( \sigma_{IoU} \mathcal{L}_{cls} + L_{reg} \right) + \frac{1}{N_{neg}} \sum_{l,t} \mathbb{1}_{\{c_t^l = 0\}} \mathcal{L}_{cls}, \tag{8}$$

where $N_{pos}$ and $N_{neg}$ are the numbers of positive and negative samples, respectively; $\mathbb{1}c_t^l > 0$ is an indicator function that equals 1 if $c_t^l > 0$ (positive sample) and 0 otherwise; $\sigma_{IoU}$ is the temporal Intersection over Union (IoU) between the predicted segment and the ground truth, serving as a weighting factor; $\mathcal{L}_{cls}$ is the classification loss, implemented as the focal loss Ross & Dollár (2017); and $\mathcal{L}_{reg}$ is the regression loss, implemented as the IoU loss Rezatofighi et al. (2019). The weighting factor $\sigma_{IoU}$ emphasizes predictions with higher temporal IoU, ensuring that higher-quality predictions contribute more significantly during training. Positive samples are determined

using center sampling, where instants near the center of an action instance are labeled as positive, and others are considered negative.

**Anomaly Detection Loss $\mathcal{L}_{AD}$ (MGFN Chen et al. (2023)):**

The full MGFN loss function is defined as:

$$\mathcal{L}_{AD} = \mathcal{L}_{sce} + \lambda_1 \mathcal{L}_{ts} + \lambda_2 \mathcal{L}_{sp} + \lambda_3 \mathcal{L}_{mc}, \tag{9}$$

where $\lambda_1 = \lambda_2 = 1$ and $\lambda_3 = 0.001$. The base loss $\mathcal{L}_{sce}$ is the standard sigmoid cross-entropy loss:

$$\mathcal{L}_{sce} = -y\log(s^{i,j}) - (1-y)\log(1 - s^{i,j}), \tag{10}$$

with $y$ as the video-level label ($y = 1$ for anomaly, $y = 0$ for normal) and $s^{i,j}$ as the computed anomaly score for frame $i$ in segment $j$. Following Sultani et al. (2018), it incorporate a temporal smoothness term $\mathcal{L}_{ts}$ and a sparsity term $\mathcal{L}_{sp}$:

$$\mathcal{L}_{ts} = \sum_{i=1}^{n-1} \left( f(V_a^i) - f(V_a^{i+1}) \right)^2, \tag{11}$$

$$\mathcal{L}_{sp} = \sum_{i=1}^{n} f(V_a^i), \tag{12}$$

where $f(V_a^i)$ represents the extracted features for segment $i$ of an anomalous video $V_a$. These terms encourage smooth transitions between sequential segments and promote sparsity in detections.

MGFN introduces a feature amplification mechanism and a magnitude contrastive loss $\mathcal{L}_{mc}$ to enhance feature separability within and between videos, formulated as:

$$\mathcal{L}_{mc} = \sum_{p,q=0}^{B/2} (1-l)(D(M_n^p, M_n^q)) + \sum_{u,v=B/2}^{B} (1-l)(D$$
$$(M_a^u, M_a^v)) + \sum_{p=0}^{B/2} \sum_{u=B/2}^{B} l(Margin - D(M_n^p, M_a^u)), \tag{13}$$

where $B$ is the batch size, $M$ denotes the feature magnitude of the corresponding segment, $l$ is an indicator function, and $D(\cdot,\cdot)$ is a distance function. Refer to Chen et al. (2023) for more details.

## C  ADDITIONAL EXPERIMENTS

**Different Architectures for Anonymizing Adapter Module (AAM):** Our ablation study evaluates different AAM architectures in Table 10, with the baseline showing standard performance without anonymization. The multi-layer perception (MLP) adapter demonstrates moderate privacy enhancement, particularly with increased capacity, while nearly maintaining utility performance. However, the self-attention-based module is superior across the board, finely balancing privacy and utility, making it our Anonymizing Adapter Module of choice. The difference between the encoder having 3 and 5 layers is negligible, as performance appears to plateau with the larger capacity. As such, for more efficient compute without sacrificing performance, we adopt the 3 encoder layer self-attention AAM for the majority of experiments. Self-attention's efficacy is likely due to its ability to prioritize crucial features for anonymization, refining the privacy preservation process.

**Relative weightage of latent consistency objective:** To further investigate the importance of latent consistency loss, we consider varying weights w.r.t. the overall training objective in Table 11. Since we want to ensure generalization across unseen tasks, action recognition is the only training utility task in this experiment. We found more solid support that with increasing the weightage of the latent consistency loss, performance maintains on the action-related utility, however, it significantly increases performance on the unseen anomaly detection task.

Table 10: Ablation for different AAM architectures. Self-attention beats out standard MLPs.

| Anonymizer Architecture | VISPR cMAP ($\downarrow$) | K400 Acc. ($\uparrow$) | UCF Cr. AUC ($\uparrow$) | THUM14 mAP ($\uparrow$) |
|---|---|---|---|---|
| None | 70.47 | 74.86 | 85.79 | 60.82 |
| MLP (1 layer) | 67.51 | 73.39 | 82.13 | 59.42 |
| MLP (3 layers) | 62.92 | 64.84 | 79.63 | 54.60 |
| MLP (5 layers) | 61.92 | 70.03 | 83.47 | 57.34 |
| Self-Attn (1 layer) | 50.59 | 72.57 | 82.12 | 58.17 |
| Self-Attn (3 layers) | 49.92 | **74.23** | **84.33** | **60.50** |
| Self-Attn (5 layers) | **48.56** | 74.08 | 83.54 | 57.46 |

Table 11: Ablation for weight of $\mathcal{L}_{LC}$.

| $\omega_{LC}$ | VISPR cMAP ($\downarrow$) | HMDB51 Top1 Acc. ($\uparrow$) | UCF Crime AUC (%) ($\uparrow$) |
|---|---|---|---|
| 0 | 51.7 | 72.88 | 65.62 |
| 1 | 48.96 | 73.27 | 72.19 |
| 10 | 52.5 | 73.4 | 72.58 |
| 100 | **54.35** | **73.92** | **84.52** |
| 1000 | 59.2 | 73.33 | 83.57 |

**Ablation with Attention head counts in AAM:** We show here in Table 12 the effect of changing the number of heads in the MHSA layer of our transformer based AAM. The performance for each variation was very similar, with the middle 8 heads beating out the other variations, providing a solid tradeoff for compute and performance. Our default experiment setup utilizes 8 MHSA heads.

Table 12: Ablation with different number of MHSA Heads.

| Num MHSA Heads | VISPR cMAP ($\downarrow$) | HMDB51 Top1 Acc. ($\uparrow$) | UCF Crime AUC (%) ($\uparrow$) |
|---|---|---|---|
| 4 | 54.78 | 73.79 | 83.73 |
| 8 | **54.35** | **73.92** | **84.52** |
| 16 | 56.74 | 73.73 | 83.99 |

Figure 3 follows Singh et al. (2022) to plot privacy-utility curve (NHV=0.6833) using PA-HMDB test set. Varying task weights leads to controllable trade-off curve.

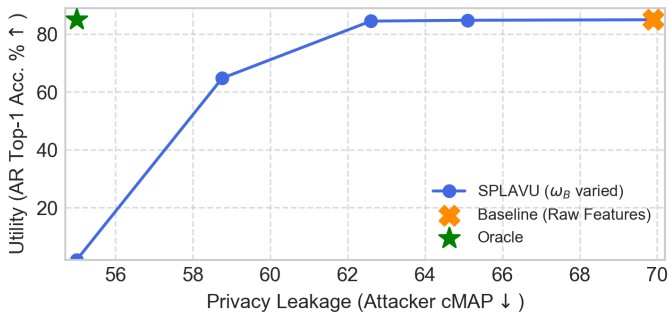

Figure 3: Privacy-utility trade-off on PA-HMDB51. Privacy measured by attacker cMAP ($\downarrow$), utility by AR acc. ($\uparrow$). Different points show varied privacy/utility weights $\omega_B, \omega_T$. SPLAVU achieves a favorable trade-off.

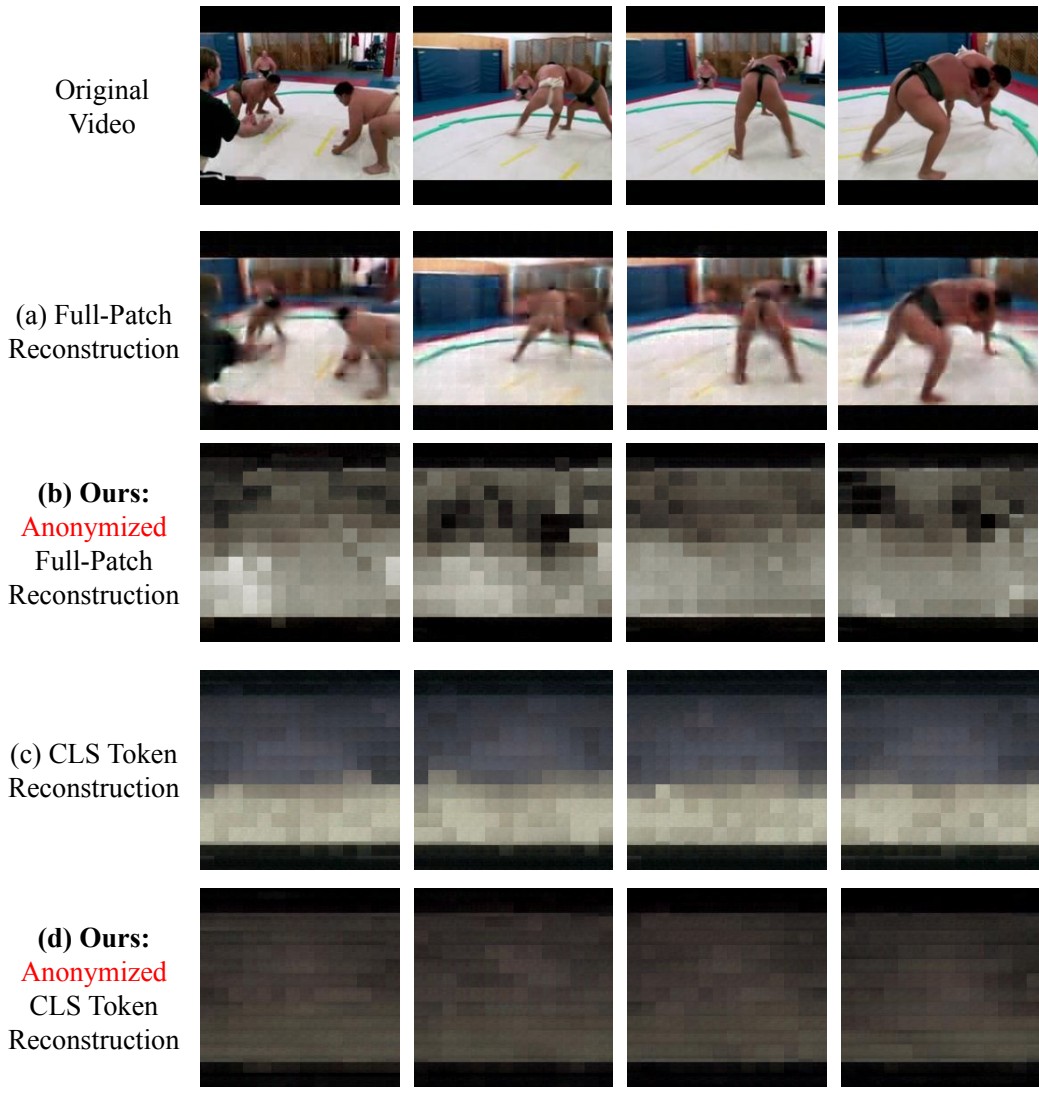

Figure 4: Qualitative reconstruction analysis using the VideoMAE decoder. We evaluate four input configurations: (a) reconstruction from non-anonymized full patch tokens, (b) reconstruction from patch tokens after applying our $f_A$, (c) reconstruction using repeated CLS tokens, and (d) reconstruction using anonymized CLS tokens. As shown, meaningful spatial structure is recoverable only when full patch tokens are provided; anonymized patches and CLS-only variants fail to reconstruct identifiable content.

Table 13: UCF101 retrieval results (query: test set, gallery: train set). Evaluation uses Recall@1, Recall@5, Recall@10, and mAP. Our anonymizer faithfully maintains retrieval performance.

| Method | R@1 | R@5 | R@10 | mAP |
|---|---|---|---|---|
| Baseline (Raw Videos) | 95.3 | 97.6 | 98.7 | 84.9 |
| Ours | 94.9 | 97.6 | 98.6 | 84.7 |

**Video Retrieval:** Table 13 shows UCF101 retrieval performance using an anonymizer trained solely on Kinetics400, demonstrating cross-dataset and cross-task generalization without any retrieval-specific supervision.

**Reconstruction Attack Mitigation:** Figure 4 illustrates our reconstruction test protocol, designed to evaluate whether our approach mitigates reconstruction-based attacks and if CLS embeddings contain sufficient information for pixel-space recovery. Using the official VideoMAE decoder, we compare four conditions: (a) reconstruction from the full set of patch tokens, which recovers the scene faithfully; (b) reconstruction after applying our $f_A$ to the patchwise tokens, which severely disrupts spatial content; (c) reconstruction when the decoder is retrained to operate on a repeated CLS token, which produces only faded scenes; and (d) reconstruction from our anonymized CLS embeddings, which similarly fails to yield meaningful structure. Each setting finetunes the decoder for 10 epochs on the UCF101 training set, ensuring fair comparison. Impressively, despite being trained to remove private attributes from CLS tokens, our method prevents reconstruction attacks from recovering the original video, even when provided patch-level tokens. We also confirm that CLS features lack the spatial detail necessary for reconstructing privacy-revealing video content.

## C.1 COMPARISON WITH OTHER PRIOR METHODS

Previous work Wu et al. (2020); Dave et al. (2022b); Fioresi et al. (2023) has already shown that the learnable anonymization techniques outperform methods such as downsampling, blurring, and blackening. Main Table 1 shows a comparison to these techniques using the I3D Carreira & Zisserman (2017) model. In Downsample-2x and Downsample-4x, the input frames have their resolution reduced by a factor of 2 ($112 \times 112$) and 4 ($56 \times 56$). In Blackening and Blurring, subjects are detected using an object detector to detect human subjects and obfuscated using the same methods as described in Wu et al. (2020); Dave et al. (2022b); Fioresi et al. (2023). We see that none of these techniques achieve an acceptable level of anonymization, and almost all reduce utility more than our SPLAVU method, further demonstrating the capability of our framework.

Table 14 shows the results of our proposed method on PA-HMDB Wu et al. (2020) compared to the baseline model on raw data.

Table 14: PA-HMDB51 results, using VideoMAE as $f_E$.

| Method | **Privacy**
cMAP ($\downarrow$) | **Action**
Top-1 Acc ($\uparrow$) |
|---|---|---|
| Baseline (Raw Videos) | 69.9 | 80.19 |
| Ours | **62.6** | **84.47** |

## C.2 ADDITIONAL EXPERIMENTS WITH LARGE FOUNDATION MODELS

Due to the low compute cost and focus on maintaining the capabilities of powerful models, our SPLAVU framework is able to scale up to the largest video foundational models currently available. Table 15 demonstrates the high privacy-utility tradeoff achieved by our method using InternVideo-H Wang et al. (2022), and Main Paper Table 1 shows results using VideoMAEv2-G Wang et al. (2023). In these experiments, action recognition performance was exactly maintained, and private attribute prediction was dropped more than for the smaller models, with only a modest reduction in temporal action detection performance.

Table 15: Performance when scaling SPLAVU up to larger models.

| Anonymization Method | Model | VISPR cMAP (↓) | HMDB51 Top 1 Acc. (↑) | UCF101 Top 1 Acc. (↑) | THUMOS14 mAP (%) (↑) |
|---|---|---|---|---|---|
| Baseline | InternVideo-H | 74.62 | 79.48 | 98.84 | 62.45 |
| Ours-HMDB51 | | 54.74 | 79.87 | – | 56.35 |
| Ours-UCF101 | | 50.29 | – | 99.21 | 53.28 |
| Baseline | VideoMAEv2-G | 75.69 | 81.05 | 97.81 | 70.09 |
| Ours-HMDB51 | | 53.39 | 80.85 | – | 65.21 |
| Ours-UCF101 | | 51.10 | – | 97.91 | 62.69 |

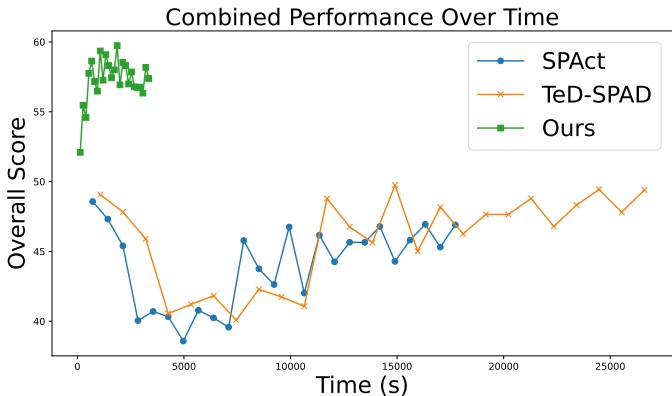

Figure 5: Graph showcasing the overall runtime and accuracy of 3 privacy-preserving methods. The x-axis shows time in seconds and the y-axis has an overall score for accuracy/privacy computed in Equation (14).

### C.2.1 TRAINING COMPUTE

One of the many benefits of our SPLAVU framework is its very low compute/training cost. Table 16 shows the overall count of trainable parameters for previous frameworks compared to our AAM. For VideoMAE-Base, our SPLAVU framework with the self-attention AAM has **88.7%** less trainable parameters when compared to existing approaches. This difference is even greater when scaling to larger models. Training less parameters can reduce the tendency to overfit on the proxy task and allow for learning an effective anonymization on limited training data (see Main Paper Table 4). Also, in federated learning, these parameters are communicated between the server and clients, so the reduced learnable parameters are useful in efficient and privacy-preserving federated learning Zhao et al. (2023); Yu et al. (2022).

Table 16: Trainable parameters for each framework/model.

| Method | Model | Trainable Params (M) |
|---|---|---|
| SPAct/TeD-SPAD | I3D | 55.2 |
| SPLAVU | | **25.6** |
| SPAct/TeD-SPAD | VideoMAE-B | 129.4 |
| SPLAVU | | **14.6** |
| SPAct/TeD-SPAD | V-JEPA | 694.2 |
| SPLAVU | | **39.8** |
| SPAct/TeD-SPAD | InternVideo-H | 675.0 |
| SPLAVU | | **39.8** |
| SPAct/TeD-SPAD | VideoMAEv2-G | 1055.5 |
| SPLAVU | | **48.0** |

Table 17: Results comparison between AAM trained on HMDB51 using input videos vs. precomputed features. Experiment was done using VideoMAE-B model.

| Training Data | PAP VISPR | AR HMDB51 | TAD T14 | AD UCF-Cr. | Training Time (min) |
|---|---|---|---|---|---|
| Raw Videos | **50.59** | **75.10** | **58.15** | 82.71 | 185.3 |
| Precom. Feats | 54.35 | 73.92 | 56.50 | **84.52** | **4.0+1.4** |

The efficiency of our method is further demonstrated using Figure 5. In this instance, our method did not make use of precomputed features, yet it still completed $\approx$**3.5x** faster than the next fastest method. The combined accuracy/privacy metric is simply defined as follows:

$$y_t = (acc_t + (1 - priv_t)) * 0.5, \tag{14}$$

where $t$ is the current time, $y_t$ is the performance score, and $acc_t$ and $priv_t$ are the top-1 accuracy scores and privacy prediction score using the current $f_A$ model, respectively. Privacy is inverted as a lower private attribute prediction score is considered better. Each method was trained for 50 epochs using the same hyperparameters. The SPLAVU latent anonymization framework achieves a higher, more stable performance at only a fraction of the runtime when compared to input-based methods.

### C.2.2 PRECOMPUTING FEATURE EMBEDDINGS

Since we are using a completely frozen video encoder model $f_E$, the latent feature embeddings can be precomputed for a much faster training process. In this case, only validation augmentations are used, and each video clip is only ran through the model forward pass once. There is flexibility in clip choice and skip rate. In this work, we opt for a simple skip rate of 1 (consecutive frames), and take all non-overlapping sequential clips for each video. The computed embeddings are saved for each video, and a random clip is sampled during training time. The same evenly-spaced 5 video clips are used for validation. Table 17 shows a comparison between using the raw videos and precomputed features. Due to the use of weak augmentations in the raw videos, we see an improvement over using the precomputed. However, using the precomputed features only requires a single forward pass over the dataset, which takes 4 minutes (HMDB51), then only **1.4** minutes for training.

## D  TRAINING ALGORITHM

Algorithm 1 formalizes the SPLAVU workflow notation. We consider anonymizer $f_A$ and task heads $f_{T_{AR}}$, $f_{T_{TAD}}$, and $f_{T_{AD}}$ for the anonymization training and $f_{AR}$, $f_{TAD}$, and $f_{wsad}$ for downstream tasks. In order, these models are parameterized by $\theta_A$, $\theta_{T_{AR}}$, $\theta_{T_{TAD}}$, $\theta_{T_{AD}}$, $\theta_{AR}$, $\theta_{TAD}$, and $\theta_{wsad}$. $\mathbb{D}_{AR}$, $\mathbb{D}_{TAD}$, and $\mathbb{D}_{wsad}$ are all used in the proxy anonymization process, then also for the downstream task evaluation. The downstream $\mathbb{D}_{AR}$ may be the same or different from during the anonymization process.

---

**Algorithm 1:** SPLAVU Framework

---

1    **Anonymization Training**

2    **Inputs**:

3      *Datasets:* $\mathbb{D}_{AR}, \mathbb{D}_{TAD}, \mathbb{D}_{wsad}$

4      *# of Epochs: anon_epochs*

5      *Learning Rates:* $\alpha_A, \alpha_{AR}, \alpha_{TAD}, \alpha_{AD}$

6      *Hyperparameters:* $\omega_A, \omega_T, \omega_B, \omega_{LC}, \omega_{AR}, \omega_{TAD}, \omega_{AD}$

7    **Output**: $\theta_A, \theta_{T_{AR}}, \theta_{T_{TAD}}, \theta_{TAD}$

---

8    Model Initialization:

9    Initialize $f_E$ with Kinetics400 weights Carreira & Zisserman (2017);

10    Initialize $\theta_A \leftarrow \theta_A - \alpha_A \nabla_{\theta_A}(\mathcal{L}_{L1}(\theta_A))$

---

11    Multitask Anonymization Training:

12    **for** $e_0 \leftarrow 1$ **to** $anon\_epochs$ **do**

13      $\theta_A \leftarrow \theta_A - \alpha_A \nabla_{\theta_A}(\omega_{LC}\mathcal{L}_{LC}(\theta_A) + \omega_T \mathcal{L}_{T^*}(\theta_A, \theta_{T_{AR}}, \theta_{T_{TAD}}, \theta_{TAD}) - \omega_B L_B(\theta_A))$

       $\theta_{T_{AR}} \leftarrow \theta_{T_{AR}} - \alpha_{AR} \nabla_{\theta_{T_{AR}}}(\mathcal{L}_{AR}(\theta_{T_{AR}}, \theta_A)),$

       $\theta_{T_{TAD}} \leftarrow \theta_{T_{TAD}} - \alpha_{TAD} \nabla_{\theta_{T_{TAD}}}(\mathcal{L}_{TAD}(\theta_{T_{TAD}}, \theta_A)),$

       $\theta_{TAD} \leftarrow \theta_{TAD} - \alpha_{AD} \nabla_{\theta_{TAD}}(\mathcal{L}_{AD}(\theta_{TAD}, \theta_A)),$

14    **end**

---

15    **Downstream Tasks Evaluation**

---

16    **Inputs**:

17      *Datasets:* $\mathbb{D}_{AR}, \mathbb{D}_{AD}, \mathbb{D}_{TAD}$

18      *# of Epochs: reco_epochs, anomaly_epochs, tad_epochs*

19      *Learning Rates:* $\alpha_{AR}, \alpha_{wsad}, \alpha_{TAD}$

20    **Output**: $\theta_{AR}, \theta_{wsad}, \theta_{TAD}$

---

21    Privacy-Preserved Action Recognition Training:

22    **for** $e_0 \leftarrow 1$ **to** $reco\_epochs$ **do**

23      $\theta_{AR} \leftarrow \theta_{AR} - \alpha_{AR} \nabla_{\theta_{AR}}(\mathcal{L}_T(\theta_{AR}, \theta_A)),$

24    **end**

---

25    Feature Extraction on $\mathbb{D}_{AD}$:

26    $\mathbb{F}_{AD} = \{ f_A(f_E(X^{(i)})) \mid \forall X^{(i)} \in \mathbb{D}_{AD} \}$

---

27    Privacy-Preserved Weakly-Supervised Anomaly Detection (WSAD) Training:

28    **for** $e_0 \leftarrow 1$ **to** $anomaly\_epochs$ **do**

29      $\theta_{wsad} \leftarrow \theta_{wsad} - \alpha_{wsad} \nabla_{\theta_{wsad}}(L_{wsad}(\theta_{wsad}, \mathbb{F}_{AD}))$

30    **end**

---

31    Feature Extraction on $\mathbb{D}_{TAD}$:

32    $\mathbb{F}_{TAD} = \{ f_A(f_E(X^{(i)})) \mid \forall X^{(i)} \in \mathbb{D}_{TAD} \}$

---

33    Privacy-Preserved Temporal Action Detection (TAD) Training:

34    **for** $e_0 \leftarrow 1$ **to** $tad\_epochs$ **do**

35      $\theta_{TAD} \leftarrow \theta_{TAD} - \alpha_{TAD} \nabla_{\theta_{TAD}}(L_{TAD}(\theta_{TAD}, \mathbb{F}_{AD}))$

36    **end**

---

