# OpenReview forum: "Privacy Beyond Pixels: Latent Anonymization for Privacy-Preserving Video Understanding"
_ICLR.cc/2026/Conference — ICLR 2026 Poster_

### Official Review · Reviewer_dNWy · 2025-10-29

**Soundness:** 2
**Presentation:** 3
**Contribution:** 2
**Rating:** 4
**Confidence:** 5

**Summary:**

The paper proposes SPLAVU, a lightweight anonymizing adapter placed on top of a frozen video encoder to remove spatially private information from video features while preserving temporal information for downstream tasks. The method is trained with three objectives: a clip-level self-supervised privacy objective to reduce mutual information between static clips, task-specific utility losses, and a latent consistency term to preserve transferability. Experiments on action recognition, temporal detection, and anomaly detection show about a 35% drop in private-attribute predictability with minimal utility loss. Privacy is evaluated on VISPR and CASIA-B, and a bias protocol is included on NTU RGB+D and Toyota Smarthome. The approach is simple and does not require retraining the video backbone.

**Strengths:**

- The paper is well motivated. Privacy preservation in video representations is an important problem, and the community should work more on this topic.
- The paper introduces the novel idea of anonymizing latent features instead of input pixels or frames.
- The proposed anonymization adapter (AAM) is lightweight, plug-and-play, and works on top of a frozen video encoder, making the approach simple and efficient to train.
- The visual explanation in Figure 2 helps readers understand the method.
- The evaluation includes privacy, fairness, and bias analysis, which broadens the impact and shows awareness of social implications.

**Weaknesses:**

- The privacy evaluation is not clear. It appears that the authors train a separate classifier on anonymized features f_A(f_E(x)) for VISPR, but the implementation details (e.g., classifier type, training schedule, multi-label setting, etc) are missing. Clarifying this would help assess reproducibility and fairness when comparing to prior privacy-preserving frameworks that integrate the classifier during training.

- The paper enforces a latent-consistency loss to keep anonymized features close to the encoder's latent space, aiming for generalization to unseen tasks. However, this design could also make the anonymized representations vulnerable to feature inversion or reconstruction attacks, since $h'_t=f_A(f_E(x_t))$ may remain close to $h_t=f_E(x_t)$. An attacker with access to the frozen encoder or a decoder trained on its features could potentially approximate the original frames. Specifically, an adversary could learn an approximate inverse $f_E^{-1}$ or a regression model $g: h'_t \rightarrow x_t$ trained on pairs $(h_t, x_t)$ or $(h'_t, x_t)$, enabling recovery of frames from anonymized features. A discussion of this risk would strengthen the paper's privacy claims.

- The paper claims generalization to "unseen tasks," yet the evaluated ones (action recognition, temporal action detection, and anomaly detection) are all closely related motion-centric problems. Assessing the method on more diverse downstream video understanding tasks, such as video retrieval, video object segmentation, video captioning, or person re-identification, would provide stronger and more convincing evidence of true task-level generalization.

- Unlike prior works such as SPAct, this paper does not include an explicit privacy classifier or adversarial optimization loop. While this design choice simplifies training, it also weakens the link between the training objective and actual privacy protection. The self-supervised "privacy budget" loss acts only as an indirect proxy for removing spatial identity information and may not guarantee robustness. It would also be helpful to visualize the input to the privacy classifier model.

- Since the privacy and utility objectives are inherently competing, I wonder if the joint optimization of $f_A$ and $f_T^{*}$, considering $L_B$ and $L_{LC}$, remains stable.

- The ablation on the latent-consistency (cycle-consistency) loss is evaluated only on HMDB51, whereas the main results (Table 1) include multiple datasets and tasks. It would be more convincing to show the same ablation across others.

- The paper fixes the clip sampling and length configuration but does not analyze its impact on privacy or utility. Since privacy leakage and temporal consistency can depend strongly on clip duration and sampling strategy, an ablation on clip length or selection would help clarify whether the proposed method's gains generalize across different temporal windows.

- The structure could be improved for clarity. While $f_A$ appears in the equations of Section 3.1, its definition is only given in Section 3.2. Introducing it earlier would make the methodological flow easier to follow.

- The notation is not clearly defined and is sometimes hard to follow. For example, $T^n$ is used in the loss formulation but never explicitly defined; it's unclear whether n indexes tasks, samples, or temporal segments.

**Questions:**

My questions are based on the weakness above. Specifically:
1. Can the authors please clarify the implementation details of the privacy classifier and provide more information on how privacy was evaluated?
2. Do the authors evaluate the robustness of their method against feature inversion or reconstruction attacks? Can the anonymized features $h'_t$ be used to approximate or reconstruct the original frames $x_t$?
3. Does the proposed method actually generalize to unseen tasks such as video retrieval, video object segmentation, captioning, or re-identification?
4. If there is no adversarial classifier or adversarial optimization during training, how do the authors ensure that the optimization process is actually guided to remove privacy-related attributes rather than destroying useful features? Simply reducing mutual information between frame features does not seem to be a strong proxy for privacy removal.
5. How does the input of the privacy classifier look like?
6. How stable is the proposed approach? Since the privacy and utility objectives are inherently competing, I wonder wheter the joint optimization of $f_A$ and $f_T^{*}$, considering $L_B$ and $L_{LC}$, remains stable during training.
7. Why is the ablation on the latent-consistency loss reported only on HMDB51? Could you extend it to another dataset shown in Table 1?
8. Did the authors analyze how clip length or sampling strategy affects privacy and utility? Would longer or shorter clips change the results?

---

> ### Author Response · Authors · 2025-11-22
> **Response to Reviewer dNWy (1/3)**
>
> Thank you for your careful evaluation, for highlighting the strengths of our approach, and for raising several insightful questions.
>
> >"Can the authors please clarify the implementation details of the privacy classifier and provide more information on how privacy was evaluated? How does the input of the privacy classifier look like?"
>
> We follow the VISPR evaluation protocol of SPAct [1], differing only in that our anonymizer operates in feature space. VISPR images are repeated 16 times to form a “static clip,” which is passed through the backbone encoder $f_E$ and anonymized via $f_A$. A 2-layer MLP maps the anonymized embedding to the 7 VISPR attribute logits. This classifier is trained with AdamW (learning rate $1\times10^{-3}$) using a BCEWithLogits loss for 100 epochs, evaluating every 5 epochs and reporting the best validation cMAP. For the VP-HMDB51 and VP-UCF101 privacy evaluations, we use standard temporal clips; their embeddings are fed to an identical attribute-classifier head.
>
> >"Do the authors evaluate the robustness of their method against feature inversion or reconstruction attacks? Can the anonymized features $h_t'$ be used to approximate or reconstruct the original frames $x_t$?"
>
> Our method operates solely on the CLS embedding used in tasks such as retrieval and those evaluated in the paper; it does not modify or expose patch-level features. When only CLS embeddings are stored—as is typical in deployed retrieval systems—reconstruction of the original frames is not feasible. An adversary can still attempt to recover the *non-anonymized* embedding, so we evaluate both black and white-box inversion attacks (Table 1). In the *black-box* setting, the attacker observes input–output pairs of the anonymizer and trains a separate inversion model where no gradients through $f_A$ are available. In the *white-box* setting, the attacker has full access to $f_A$ and instead performs gradient-based reconstruction by optimizing a synthetic embedding to match the observed anonymized output. The black-box inversion network can recover action-classification performance but does not improve VISPR attribute prediction beyond random chance, indicating limited leakage of private attributes. White-box attacks remain effective, though they assume full access to model parameters, which is uncommon in practical deployments. Improving robustness in this setting is an important direction for future work.
>
> **Table 1:** Feature inversion attack results. Our model remains robust under black-box attacks: action accuracy can be partially recovered, but private attribute prediction stays at chance. Random-chance baselines are included for context.
> |Method|VP-HMDB51 Acc.$\uparrow$|VP-HMDB51 cMAP$\downarrow$|VP-UCF101 Acc.$\uparrow$|VP-UCF101 cMAP$\downarrow$|
> |-|:-:|:-:|:-:|:-:|
> |Raw Features|72.6|76.4|96.8|75.9|
> |Ours|72.1|70.5|96.8|69.6|
> |Random Chance|--|58.3|--|58.7|
> |Black Box Inversion|67.3|58.8|96.8|58.4|
> |White Box Inversion|68.0|73.7|94.5|72.6|
>
> >"Does the proposed method actually generalize to unseen tasks such as video retrieval, video object segmentation, captioning, or re-identification?"
>
> We agree that broader task diversity would strengthen the evidence for task-level generalization. We focus on tasks directly utilizing the compressed CLS token embedding rather than dense tasks that require patch tokens. We will clarify this limitation in future revisions. That said, the anonymizer does transfer to tasks it never sees during training. For example, Table 2 shows UCF101 retrieval performance using an anonymizer trained solely on Kinetics400, demonstrating cross-dataset and cross-task generalization without any retrieval-specific supervision.
>
> **Table 2:** UCF101 retrieval results (query: test set, gallery: train set). Evaluation uses Recall@1, Recall@5, Recall@10, and mAP.
> |Method|R@1|R@5|R@10|mAP|
> |-|:-:|:-:|:-:|:-:|
> |Raw Features|95.3|97.6|98.7|84.9|
> |Ours - K400|94.9|97.6|98.6|84.7|

---

> > ### Comment · Reviewer_dNWy · 2025-11-28
> > **Follow-up questions**
> >
> > Dear authors,
> >
> > I really appreciate your careful response and the additional experiments. I have some follow-up questions.
> >
> > In your response, you state that:
> >
> > > "Our method operates solely on the CLS embedding … and does not modify or expose patch-level features."
> >
> > However, I could not find this described in the paper. In particular, Figure 2 shows the anonymization module $f_A$ operating on the full clip-level feature sequence output by the video encoder $f_E$, not just a single CLS token, and Eq. (6) defines the latent consistency loss over the entire feature map $f_E(x)$. Could you please clarify where in the paper it is stated that the proposed method operates only on the CLS embedding?
> >
> > Similarly, in your response, you mention that the attacker attempts to recover the "non-anonymized embedding." To clarify: does this refer to the original encoder output $h_t = f_E(x_t)$?
> >
> > If so, I am concerned that the latent consistency loss (Eq. 6) explicitly encourages $h'_t = f_A(h_t) \approx h_t$.
> >
> > Given this, an attacker could train a regression network $g: h'_t \rightarrow h_t$ using observed anonymized embeddings, recover an estimate of $h_t$, and then apply existing feature-inversion or reconstruction techniques to approximate the original frames $x_t$.
> >
> > Have you evaluated (or can you comment on) the feasibility of such an attack? In particular, how robust is SPLAVU to inversion attacks that attempt to map $h'_t \to h_t \to x_t$, especially given the latent consistency constraint in Eq. (6)? Given the time constraints, I am not expecting you to run new experiments, but I would be very interested to hear your thoughts on this threat model.
> >
> > On the other hand, you wrote in your response:
> >
> > > "We agree that broader task diversity would strengthen the evidence for task-level generalization. We focus on tasks directly utilizing the compressed CLS token embedding rather than dense tasks that require patch tokens. We will clarify this limitation in future revisions."
> >
> > However, I could not locate this limitation in the revised manuscript. Could you point to where this is stated?

---

> > > ### Author Response · Authors · 2025-11-28
> > >
> > > Thank you for bringing these points up, it is important to clearly define the notation and avoid confusion.
> > >
> > > >"Could you please clarify where in the paper it is stated that the proposed method operates only on the CLS embedding?"
> > >
> > > We apologize for this notational ambiguity. Our intention was to use $f_E(x_t)$ as a unified encoder representation across architectures, where the downstream anonymizer $f_A$ always operates on the final clip-level embedding (CLS for transformers, pooled vector for CNNs). For clarity, clip-level here refers to the fact that full spatio-temporal information is compressed into the global embedding.
> > >
> > > In standard practice, transformer-based video encoders (e.g., VideoMAE, ViViT) produce a sequence of tokens prepended by a special CLS token: $f_E(x_t) = [h_t^{CLS};h_t^0;\ldots;h_t^N]$, where $h_t^{\text{CLS}}$ is the global classification token and $h_t^{0},\ldots,h_t^{N}$ denote patchwise spatio-temporal tokens. In CNN-based video encoders (e.g., I3D, R3D), the encoder processes the input clip through a hierarchy of spatiotemporal convolutional blocks, producing a feature tensor $f_E(x_t) = \mathbf{F}_t \in \mathbb{R}^{T' \times H' \times W' \times C},$ which is then aggregated into a global clip-level representation via spatiotemporal average pooling: $h_t^{\text{pool}} = \text{AvgPool}(\mathbf{F}_t) \in \mathbb{R}^{C}.$ This pooled feature $h_t^{\text{pool}}$ serves as the analog to the CLS token in transformer-based encoders. Because our anonymizer operates on the unified notion of a single clip-level representation, either the CLS token or the globally pooled CNN feature, we simplify notation by writing $f_E(x_t)$ to refer to this global embedding regardless of encoder type.
> > >
> > > We now explicitly state this in various places, including in Section 3.1 and 3.2, the Figure 2 caption, and a dedicated Appendix subsection B.1. As for Figure 2, the intention was to show that $f_A$ acts on the output of $f_E$ which is utilized for the shown downstream tasks; either the global CLS embedding or the output of the average pooling layer. Assuming that the writing has made this clear, explicitly visualizing this would complicate the figure and may further confuse readers. We have updated the caption to directly mention this.
> > >
> > > As for addressing the lack of dense task handling:
> > > >"However, I could not locate this limitation in the revised manuscript. Could you point to where this is stated?"
> > >
> > > This is added to the limitations and highlighted in blue text. Additionally, earlier text now clarifies task usage of a single global clip embedding (Section 3.2).
> > >
> > >
> > > >"Given this, an attacker could train a regression network $g : h_t' \rightarrow h_t$ using observed anonymized embeddings, recover an estimate of $h_t$, and then apply existing feature-inversion or reconstruction techniques to approximate the original frames $x_t$.
> > >
> > > We believe that our earlier response (in Response to Reviewer dNWy (1/3), black and white-box attacks, Table 1) addresses most of this follow-up question. There, in the black-box attack setting, we explicitly train a regression network $g : h_t' \rightarrow h_t$. Notably, we find that $g$ can recover action recognition performance but does *not* improve private attribute prediction beyond random chance. This indicates that the predicted $h_t$ lacks private information, meaning any reconstructed $x_t$ will not reliably reveal private attributes. While we did not explicitly continue the chain to complete $h_t' \rightarrow h_t \rightarrow x_t$, the resistance to $h_t' \rightarrow h_t$ is highly encouraging.
> > >
> > > Although Eq. 6 encourages the anonymized embedding to remain close to the original one, in practice it functions as a regularizer rather than a direct reconstruction objective. The latent consistency loss prevents the anonymizer from drifting into task-specific or degenerate feature spaces, but it does not override the privacy objective and retain private attribute knowledge (at the correct weightage, see Appendix Table 9). The inversion attack results in our previous response Table 1 further confirm that this regularization does not make the embedding more vulnerable to reconstruction attacks.

---

> ### Author Response · Authors · 2025-11-22
> **Response to Reviewer dNWy (2/3)**
>
> >"If there is no adversarial classifier or adversarial optimization during training, how do the authors ensure that the optimization process is actually guided to remove privacy-related attributes rather than destroying useful features? Simply reducing mutual information between frame features does not seem to be a strong proxy for privacy removal."
>
> Our approach intentionally avoids using private attribute labels during training. While this weakens the direct alignment between the training loss and privacy removal, it offers a practical advantage: the method scales without requiring sensitive attribute annotations. To validate that our proxy objective is still effective, we report results on VP-HMDB51 and VP-UCF101 [2], which directly measure temporal private-attribute leakage (Table 3). We also compare against a variant trained with an explicit attribute classifier. The performance gap between the two settings is minimal, suggesting that our label-free proxy objective achieves comparable privacy removal without relying on direct attribute supervision. Additionally, our gender bias protocol experiments indicate reduced ability to perceive gender.
>
> **Table 3:** Privacy evaluation on video-level private attribute benchmarks VP-HMDB51 and VP-UCF101 [2]. Random chance cMAP is relatively high for both datasets (58.3 and 58.7, respectively).
> |Method|VP-HMDB51 Acc.$\uparrow$|VP-HMDB51 cMAP$\downarrow$|VP-UCF101 Acc.$\uparrow$|VP-UCF101 cMAP$\downarrow$|
> |-|:-:|:-:|:-:|:-:|
> |Raw Features|72.6|76.4|96.8|75.9|
> |Ours w/ Attribute Classifier|72.4|70.4|96.5|69.5|
> |Ours w/ SSL|72.1|70.5|96.8|69.6|
>
> >"How stable is the proposed approach? Since the privacy and utility objectives are inherently competing, I wonder wheter the joint optimization of $f_A$ and $f_T^*$, considering $L_B$ and $L_{LC}$, remains stable during training."
>
> We are more stable than stepwise approaches such as SPAct [1]. Operating in latent space allows larger batch sizes and smaller networks, which keep joint optimization stable. In our testing, the difference between alternating and joint updates is minimal. Figure 4 (Appendix C) illustrates this: the green line shows relatively smooth training dynamics when $f_A$ and $f_{T^*}$ are optimized jointly. While some adversarial oscillation remains, the overall learning process is noticeably more stable than in prior methods.
>
> >"Why is the ablation on the latent-consistency loss reported only on HMDB51? Could you extend it to another dataset shown in Table 1?"
>
> We originally reported the latent-consistency ablation on HMDB51 because it is a small dataset with well-curated labels, making it suitable for evaluating generalization effects. We agree that extending this analysis improves completeness. Table 4 now includes corresponding ablations on Kinetics400, UCF-Crime, and THUMOS14. Interestingly, action recognition is less sensitive to variation, and performance is recoverable under different-task training.
>
> **Table 4:** Latent-consistency ablation when training $f_A$ on different datasets (“Ours–Dataset”). In all cases, removing the latent-consistency loss leads to overfitting to the training task, while adding it recovers strong cross-dataset generalization.
> |Configuration|Use $\mathcal{L}_{LC}$|VISPR$\downarrow$|K400|UCF-Crime|THUMOS14|
> |-|:-:|:-:|:-:|:-:|:-:|
> |Raw Videos|n/a|70.5|74.9|85.8|60.8|
> |Ours - K400|X|44.3|74.7|69.7|17.2|
> |Ours - K400|$\checkmark$|52.6|74.7|83.5|56.5|
> |Ours - UCF-Crime|X|47.8|73.8|85.2|13.1|
> |Ours - UCF-Crime|$\checkmark$|49.3|73.5|84.6|57.3|
> |Ours - THUMOS14|X|43.0|73.2|71.8|58.2|
> |Ours - THUMOS14|$\checkmark$|50.2|73.9|83.7|58.8|

---

> ### Author Response · Authors · 2025-11-22
> **Response to Reviewer dNWy (3/3)**
>
> >"Did the authors analyze how clip length or sampling strategy affects privacy and utility? Would longer or shorter clips change the results?"
>
> See Table 5 below for an ablation on clip length, where we vary the skip rate to adjust the effective temporal window (approximate durations assume 25 FPS). Note that the robustness of downstream frozen backbones and task models to changes in temporal window size also affects performance. Table 6 reports an ablation on clip sampling strategy: our default is random sampling, and we compare against fixed sampling from the first frame (“start”) and last possible window (“end”).
>
> **Table 5:** Ablation on temporal-window size. Increasing the skip rate enlarges the effective temporal window. Approximate durations assume 25 FPS.
> |Method|Skip Rate (Approx. Length)|HMDB51|UCF-Crime|THUMOS14|
> |-|:-:|:-:|:-:|:-:|
> |Raw Videos|1 (.6s)|74.2|85.8|60.8|
> |Ours|1 (.6s)|73.9|85.3|56.5|
> |Raw Videos|2 (1.3s)|74.1|82.8|57.2|
> |Ours|2 (1.3s)|73.5|81.2|56.3|
> |Raw Videos|4 (2.6s)|75.1|84.1|49.3|
> |Ours|4 (2.6s)|74.9|82.7|47.9|
> |Raw Videos|8 (5.1s)|75.2|82.2|33.8|
> |Ours|8 (5.1s)|74.9|81.5|32.4|
>
> **Table 6:** Clip-sampling strategy ablation during action-recognition training of $f_A$.
> |Sampling Strategy|VISPR$\downarrow$|HMDB51|UCF-Crime|THUMOS14|
> |-|:-:|:-:|:-:|:-:|
> |Random|54.4|73.9|84.5|56.5|
> |Start|55.5|63.8|81.4|53.0|
> |End|55.6|67.8|81.2|53.0|
>
> **Paper structure clarity:** Thank you for catching these, we will update the manuscript with implemented fixes. In $T^n$, n indexes tasks.
>
> ### Citations
> [1] Dave, Ishan Rajendrakumar, Chen Chen, and Mubarak Shah. "Spact: Self-supervised privacy preservation for action recognition." Proceedings of the IEEE/CVF Conference on Computer Vision and Pattern Recognition. 2022.
>
> [2] Li, Ming, et al. "Stprivacy: Spatio-temporal privacy-preserving action recognition." Proceedings of the IEEE/CVF International Conference on Computer Vision. 2023.

---

> ### Author Response · Authors · 2025-11-26
>
> >"Do the authors evaluate the robustness of their method against feature inversion or reconstruction attacks? Can the anonymized features $h_t'$ be used to approximate or reconstruct the original frames $x_t$?"
>
> To follow up on this point, we have explicitly conducted this experiment, training a decoder on $h_t'$ to reconstruct the original frames $x_t$. Impressively, despite the CLS token focus, our method is able to prevent faithful reconstruction of the original frames. Please see the updated appendix for qualitative results. Specifically, we finetune the VideoMAE decoder to attempt reconstruction of the original frames. Figure 4 (Appendix Section C, Page 19) shows four reconstruction settings: (a) reconstruction from full patch tokens, (b) reconstruction after applying our $f_A$ to patchwise tokens, \(c) reconstruction from CLS tokens, and (d) reconstruction from anonymized CLS tokens. Notably, the decoder loses reconstruction ability when $f_A$ is applied to patchwise tokens, even though $f_A$ is trained only on CLS features. It also fails to recover meaningful spatial content from CLS features, anonymized or not. These results confirm that our anonymization strongly resists video reconstruction attacks. Table 7 below contains quantitative MSE loss scores on decoded videos, further validating these qualitative results.
>
>
> **Table 7:** Reconstruction performance on the UCF101 split 1 test set, reported as MSE loss.
>
> |Method|MSE Loss$\downarrow$|
> |-|:-:|
> |(a) Raw patchwise tokens|0.0060|
> |(b) Anonymized patchwise tokens|0.0333|
> |\(c) Raw CLS token|0.0538|
> |(d) Anonymized CLS token|0.0680|

---

### Official Review · Reviewer_mpGY · 2025-10-31

**Soundness:** 3
**Presentation:** 3
**Contribution:** 3
**Rating:** 6
**Confidence:** 4

**Summary:**

This paper studies privacy preserving action recogniton under the umbralle of video foundation models. It suggests adapating the feature output of VFMs with a goal of 1) hidding the privacy information and 2) maintaining the capability of performing well on down-stream tasks. To achieve that, it introduces three objectives to be learnined together to an ANONYMIZE the latent embeddings - contrastive loss to reduce the mutal information, down-stream utiliy loss and latent space regulaization loss. Extensive experiment have been done to demonstrate the success of the method, including both appearance based privacy and motion-based privacy (e.g., gaiting).

**Strengths:**

A. The most notable strengths of the paper is the extensive experiments - testing across benchmarks and domains. The scope is comprehensive enough to prove the point. Works across multiple downstream tasks (action, temporal detection, anomaly detection) using the same anonymized features. The engineer efforts are huge without saying. Hope the code will be released for reproduction purpose.

B. The delivery of the paper is clear -well structured wording, tables and figures.

**Weaknesses:**

A. The focus on latent space, rather than pixel space, poses a challenge on interpreting the ANONYMIZED video with human-eye level intuition - Can not visually see the privacy hideout anymore. Let's consider a situation where a well-trained VFMs decoder exist (e.g., video generative model that takes VideoMAE-v2 latent embedding as input), is it possible such as generative decoder can still produce privacy revealing outputs in pixel spaces with the adapted embedding as input? This is not necessarily an attack on the paper but an open discussion chat. Wondering how would authors think about the interpretabiliy of the proposed method?

B. Table 5 shows that the L_T is very important to the recogntion performance. And the training of L_T is expensive - it requires labels, and it is concerning how much label it takes for the adapted embedding to be generalized again for a large scope of video understanding tasks, which is the original mission of VFMs. The fact that the method uses multiple downstream tasks as L_T make it hard to draw conclusion on this question. This reviwer has critical objections on this matter. It can be the case that the contrastive loss is too strong so that it destroys the useful information in the embedding space after massive data pre-training, again on the opposite direction to the VFMs. Any thoughts on how to reduce the impact of L_T training, like reduce the amount of label needed, the amount of data needed, etc,  so that the entire method is less data-hungry and by nature maintaining represenatation power with cheap buget?

**Questions:**

N/A

---

> ### Author Response · Authors · 2025-11-22
> **Response to Reviewer mpGY**
>
> We appreciate your positive remarks on our experiments and presentation, as well as the constructive concerns you highlighted.
>
> >"The focus on latent space, rather than pixel space, poses a challenge on interpreting the ANONYMIZED video with human-eye level intuition - Can not visually see the privacy hideout anymore. Let's consider a situation where a well-trained VFMs decoder exist (e.g., video generative model that takes VideoMAE-v2 latent embedding as input), is it possible such as generative decoder can still produce privacy revealing outputs in pixel spaces with the adapted embedding as input?"
>
> This is an important point, and we agree that pixel-space interpretability is a challenge when anonymization occurs in latent space. In the setting we study, only CLS embeddings are stored or exchanged, as is standard for retrieval and many downstream tasks. These embeddings do not contain the patch-level representations required by modern video decoders or VAEs. As a result, even a strong generative decoder would be unable to reconstruct meaningful pixel-space content from CLS-only features. Extending anonymization to dense patch-wise tokens is an important direction for future work. To partially address interpretability concerns, we perform a gender-bias analysis to verify that sensitive attributes are indeed suppressed (Main Paper Table 2).
>
> Additionally, we analyze a similar setting where an attacker may try and reconstruct the original embeddings to identify private attributes through black and white-box attacks. In the *black-box* setting, the attacker observes input–output pairs of the anonymizer and trains a separate inversion model where no gradients through $f_A$ are available. In the *white-box* setting, the attacker has full access to $f_A$ and instead performs gradient-based reconstruction by optimizing a synthetic embedding to match the observed anonymized output. See Table 1 below for results. Notably, in the black-box setting, the inversion network successfully recovers action classification performance, yet does not improve attribute classification performance above random chance. The model is not robust to white box attacks, which are typically unrealistic in deployment scenarios.
>
> **Table 1:** Feature inversion attack results. Our model demonstrates strong robustness to black-box attacks, where action performance is recovered but not private attribute prediction. Random chance on each dataset is included to contextualize performance.
>
> |Method|VP-HMDB51 Acc.$\uparrow$|VP-HMDB51 cMAP$\downarrow$|VP-UCF101 Acc.$\uparrow$|VP-UCF101 cMAP$\downarrow$|
> |-|:-:|:-:|:-:|:-:|
> |Raw Features|72.6|76.4|96.8|75.9|
> |Ours|72.1|70.5|96.8|69.6|
> |Random Chance|--|58.3|--|58.7|
> |Black Box Inversion|67.3|58.8|96.8|58.4|
> |White Box Inversion|68.0|73.7|94.5|72.6|
>
>
> >"Table 5 shows that the L_T is very important to the recogntion performance. And the training of L_T is expensive - it requires labels, and it is concerning how much label it takes for the adapted embedding to be generalized again for a large scope of video understanding tasks, which is the original mission of VFMs. The fact that the method uses multiple downstream tasks as L_T make it hard to draw conclusion on this question. This reviwer has critical objections on this matter. It can be the case that the contrastive loss is too strong so that it destroys the useful information in the embedding space after massive data pre-training, again on the opposite direction to the VFMs. Any thoughts on how to reduce the impact of L_T training, like reduce the amount of label needed, the amount of data needed, etc, so that the entire method is less data-hungry and by nature maintaining represenatation power with cheap buget?"
>
> Relying on utility labels is a limitation shared by most privacy-preserving methods, as supervision is needed to preserve task performance after anonymization. This concern is a key motivation for introducing our latent-consistency regularization. Rather than requiring labels for every downstream task, the latent-consistency term allows the anonymizer to maintain alignment with the pretrained backbone’s representation space. As a result, only a single set of labels is needed to learn an anonymizer that generalizes across tasks (see Main Paper Table 3). We show that training on a small dataset such as HMDB-51 (~3k video-level labels) is sufficient to learn an anonymizer that transfers effectively to much larger and more diverse datasets, including Kinetics-400 and additional downstream tasks (Main Paper Table 4). This suggests that the contrastive term does not overwhelm or overwrite the pretrained space; instead, the latent formulation preserves representation structure while removing private attributes. A promising avenue for future work is to accomplish privacy-preservation without any task labels.

---

> > ### Comment · Reviewer_mpGY · 2025-11-24
> >
> > Hi, thanks for the efforts.
> >
> > > As a result, even a strong generative decoder would be unable to reconstruct meaningful pixel-space content from CLS-only features
> >
> > Wondering how this statement is concluded. Could you please provide some cites or experimental arguments?

---

> ### Comment · Reviewer_mpGY · 2025-11-25
>
> > ... yet does not improve attribute classification performance above random chance
>
> which is a good thing for VP-UCF-101 (?) Is it sensible that Black Box Inversion beats "Ours" on VP-UCF-101 - on par on action accuracy and much lower on privacy cmap (which is more superior)?

---

> > ### Author Response · Authors · 2025-11-26
> >
> > Thanks for the quick responses, it is greatly appreciated.
> >
> > >"Wondering how this statement is concluded. Could you please provide some cites or experimental arguments?"
> >
> > While there aren't works that explicitly claim “CLS-only decoding is impossible,” the design of modern masked autoencoders strongly implies this limitation. In MAE, the decoder explicitly requires patch-level tokens: “The input to the MAE decoder is the full set of tokens consisting of (i) encoded visible patches and (ii) mask tokens” [1]. VideoMAE follows the same formulation: “A shallow decoder is placed on top of the visible tokens from the encoder and learnable mask tokens to reconstruct the image” [2]. In both cases, reconstruction depends on spatially localized patch representations, not on the global CLS token. This is further supported by recent robotics work that introduces an additional reconstruction-friendly embedding token precisely because standard representations do not allow full decoding from a single token [3]. This reinforces that current video generative decoders are not designed to take a single global token as input.
> >
> > To provide empirical evidence, we conduct a controlled experiment using the official VideoMAE decoder. We modify the decoder so that it receives only the CLS token, repeated to match the expected token count and augmented with positional embeddings, and finetune it to reconstruct frames. Qualitative results are included in the revised appendix. Figure 4 (Appendix Section C, Page 19) shows four reconstruction settings: (a) reconstruction from full patch tokens, (b) reconstruction after applying our $f_A$ to patchwise tokens, \(c) reconstruction from CLS tokens, and (d) reconstruction from anonymized CLS tokens. Notably, the decoder loses reconstruction ability when $f_A$ is applied to patchwise tokens, even though $f_A$ is trained only on CLS features. It also fails to recover meaningful spatial content from CLS features, anonymized or not. These results confirm that  our anonymization disrupts video reconstruction attacks and that CLS embeddings do not contain sufficient spatial information for pixel-level reconstruction. Table 2 below contains quantitative MSE loss scores on decoded videos, further validating these qualitative results.
> >
> >
> > **Table 2:** Reconstruction performance on the UCF101 split 1 test set, reported as MSE loss.
> >
> > |Method|MSE Loss$\downarrow$|
> > |-|:-:|
> > |(a) Raw patchwise tokens|0.0060|
> > |(b) Anonymized patchwise tokens|0.0333|
> > |\(c) Raw CLS token|0.0538|
> > |(d) Anonymized CLS token|0.0680|
> >
> >
> > >"Is it sensible that Black Box Inversion beats "Ours" on VP-UCF-101 - on par on action accuracy and much lower on privacy cmap (which is more superior)?"
> >
> > Thank you for raising this point. To clarify, the “Black Box Inversion” row is not a competing anonymization method. It is an *attack applied to our anonymized features*, where an adversary attempts to recover the original (non-anonymized) embedding and then re-run downstream classifiers on top of that recovered embedding.
> >
> > The key observation is that this attacker successfully regains action recognition performance, showing that utility can be reconstructed as intended, but gains **no improvement in private-attribute prediction**, remaining at chance level. This is precisely the desired outcome: the attack does not extract any additional privacy-relevant information beyond what our anonymized features already reveal. In other words, an adversary is strictly worse off trying to infer private attributes through this inversion attack than by using our anonymized embeddings directly, where we already demonstrate strongly reduced attribute leakage.
> >
> >
> > **Citations:**
> >
> > [1] He, Kaiming, et al. "Masked autoencoders are scalable vision learners." Proceedings of the IEEE/CVF conference on computer vision and pattern recognition. 2022.
> >
> > [2] Tong, Zhan, et al. "Videomae: Masked autoencoders are data-efficient learners for self-supervised video pre-training." Advances in neural information processing systems 35 (2022): 10078-10093.
> >
> > [3] Kim, Taekyung, et al. "Token bottleneck: One token to remember dynamics." arXiv preprint arXiv:2507.06543 (2025).

---

### Official Review · Reviewer_u3Uc · 2025-11-01

**Soundness:** 3
**Presentation:** 3
**Contribution:** 2
**Rating:** 4
**Confidence:** 4

**Summary:**

The paper proposes a privacy-preserving framework based on video foundation models by introducing a lightweight AAM at the backend of the video encoder. The AAM operates in the feature space to remove privacy-related information, and features a plug-and-play design that maintains task performance across multiple downstream tasks while achieving effective anonymization. Additionally, the method demonstrates potential in mitigating biases related to attributes such as gender.

**Strengths:**

The AAM module is designed to be lightweight and modular, enabling seamless integration with various video foundation models. It introduces minimal architectural interference to the original model and ensures efficient inference, reflecting strong modularity and scalability.

Extensive experiments are conducted on diverse datasets, including Kinetics-400 (K400), UCF101, HMDB51, ToyotaSmartHome (ToyotaSH), UCF-Crime, and THUMOS14, covering multiple downstream tasks such as action recognition, temporal action localization, and anomaly detection. This broad evaluation demonstrates the method’s generalizability and reliability, thereby strengthening the validity of the experimental conclusions.

The work introduces an assessment of gender bias within the context of video-based privacy protection, which adds social relevance and reflects forward-looking consideration of ethical implications in vision systems.

**Weaknesses:**

The design rationale of the AAM module lacks sufficient elaboration. There is inadequate theoretical justification for key design choices—such as the selection of the module’s architecture and the weighting scheme in the loss function—and insufficient ablation studies to validate their effectiveness and superiority. As a result, the optimality and robustness of the proposed design remain insufficiently supported.

Privacy preservation performance is primarily evaluated using the VISPR dataset. This evaluation approach is limited in scope and lacks validation under diverse privacy attack benchmarks or against dynamically defined sensitive attributes, which may undermine the comprehensiveness of the privacy analysis.

While the paper presents preliminary analysis on gender bias mitigation, it lacks systematic comparison with existing debiasing methods. Consequently, it is difficult to assess the relative advantages or practical efficacy of the proposed approach in addressing algorithmic bias.

**Questions:**

What motivated the choice of a 3-layer multi-head self-attention (MH Self-Attention) structure as the core component of the AAM? Were alternative lightweight architectures or different depths considered and compared through controlled ablation experiments to justify the effectiveness and necessity of this specific configuration?

On what basis were the weights assigned to the individual terms in the loss function determined? Was a systematic hyperparameter tuning procedure conducted to ensure that the selected weight combinations achieve an optimal trade-off between anonymization performance and task utility across different datasets and tasks?

In the anomaly detection task (e.g., on UCF-Crime), the reported AUC values do not reach state-of-the-art levels. Could the authors provide an analysis of potential factors contributing to this performance gap? For instance, does the anonymization process inadvertently filter out subtle but critical cues necessary for detecting anomalies?

---

> ### Author Response · Authors · 2025-11-22
> **Response to Reviewer u3Uc (1/2)**
>
> Thank you for highlighting the practical and ethical merits of our work and for pointing out areas of improvement.
>
> >"What motivated the choice of a 3-layer multi-head self-attention (MH Self-Attention) structure as the core component of the AAM? Were alternative lightweight architectures or different depths considered and compared through controlled ablation experiments to justify the effectiveness and necessity of this specific configuration?"
>
> Please see Table 8 (Appendix Section C) for an ablation of the AAM architecture. We selected the 3-layer multi-head self-attention design after comparing against shallower/deeper attention stacks and MLP-based variants. We also experimented with a lightweight LoRA-style adaptation layer in early trials, but it produced negligible privacy reduction (VISPR cMAP ≈ 70.24), so we excluded it from further evaluation. As for weighting, Table 9/Figure 3 (Appendix Section C) show that weighting can adjusted to help navigate privacy-utility tradeoff. For completeness, Table 10 (Appendix Section C) includes an ablation on the number of attention heads in each MHSA layer.
>
> >"Privacy preservation performance is primarily evaluated using the VISPR dataset. This evaluation approach is limited in scope and lacks validation under diverse privacy attack benchmarks or against dynamically defined sensitive attributes, which may undermine the comprehensiveness of the privacy analysis."
>
> We use VISPR as it is currently an established protocol in this space. To broaden the evaluation beyond frame-level attributes, we additionally include VP-HMDB51 and VP-UCF101 [2], two recent datasets annotated with *video-level private attributes*. Results for both are provided in Table 1 below. Note that while the evaluation attributes in each protocol are not strictly "dynamically defined", our method does not use knowledge of these private attributes during training. Rather, we rely on the implicit assumption that these attributes are present in static scenes. Despite not defining the attribute set during training, our method remains competitive with approaches that explicitly train on predefined sensitive attributes. This suggests that the proposed anonymization generalizes across different privacy benchmarks and is not tied to VISPR alone.
>
> **Table 1:** Privacy evaluation using video-level private attribute protocols VP-HMDB51 and VP-UCF101 [2].
> |Method|VP-HMDB51 Acc.$\uparrow$|VP-HMDB51 cMAP$\downarrow$|VP-UCF101 Acc.$\uparrow$|VP-UCF101 cMAP$\downarrow$|
> |-|:-:|:-:|:-:|:-:|
> |Raw Features|72.6|76.4|96.8|75.9|
> |Latent Anon. w/ Attribute Classifier|72.4|70.4|96.5|69.5|
> |Ours (w/ SSL)|72.1|70.5|96.8|69.6|
>
>
> >"While the paper presents preliminary analysis on gender bias mitigation, it lacks systematic comparison with existing debiasing methods. Consequently, it is difficult to assess the relative advantages or practical efficacy of the proposed approach in addressing algorithmic bias."
>
> The main purpose the bias evaluation was to highlight the intuitive finding that we *implicitly* debias by removing private information. We agree that comparison against established debiasing frameworks will provide a clearer assessment. Table 2 includes results from two representative baselines—an adversarial debiasing objective and INLP [3]. Despite not using gender labels during training, our privacy-preserving method achieves comparable reductions in gender leakage.
>
> **Table 2:** Debiasing performance comparison on NTU-Bias-F. The proposed privacy-based anonymization performs on par with debiasing-specific baselines.
>
> |Method|Male|Female|
> |-|:-:|:-:|
> |Baseline|56.2|46.8|
> |Adv. Gender Debias|50.3|48.1|
> |INLP [3]|55.8|51.3|
> |Ours|55.4|49.9|

---

> > ### Author Response · Authors · 2025-11-22
> > **Response to Reviewer u3Uc (2/2)**
> >
> > >"On what basis were the weights assigned to the individual terms in the loss function determined? Was a systematic hyperparameter tuning procedure conducted to ensure that the selected weight combinations achieve an optimal trade-off between anonymization performance and task utility across different datasets and tasks?"
> >
> > Our goal was to show that the proposed anonymization framework does not depend on extensive hyperparameter tuning. As such, we assign equal weights (=1) to each task loss term across all experiments. Despite this simple setting, our method achieves strong, SOTA privacy–utility trade-offs. A full hyperparameter sweep would yield further gains. Table 9 (Appendix Section C) includes an order-of-magnitude sensitivity sweep on $\mathcal{L}_{LC}$, showing that it needed relative upweighting to maintain generalization. Figure 3 (Appendix Section C) demonstrates that varying relative weights of $\omega_B$ and $\omega_T$ leads to a trade-off curve that can be navigated according to custom criteria.
> >
> >
> > >"In the anomaly detection task (e.g., on UCF-Crime), the reported AUC values do not reach state-of-the-art levels. Could the authors provide an analysis of potential factors contributing to this performance gap? For instance, does the anonymization process inadvertently filter out subtle but critical cues necessary for detecting anomalies?"
> >
> > For fair comparison with prior privacy-preserving methods, we follow the TeD-SPAD [4] protocol for I3D, which finetunes the backbone on UCF-101 rather than using Kinetics-400 pretrained weights. We will explicitly state this in the updated manuscript. Under this setting, the baseline itself is notably lower than the original MGFN [5] results, which also report performance using 10-crop testing instead of the single-crop protocol used here. These differences account for the apparent gap to SOTA. Regarding the anonymization process, we find no evidence that it removes cues essential for anomaly detection: when using a stronger backbone initialization (Kinetics-400 pretrained I3D), the anonymized model recovers substantially higher AUC (Table 3).
> >
> > **Table 3:** I3D task results using Kinetics-400 pretrained weights with a single crop. The observed trends are consistent with those seen across other backbone architectures.
> > |Method|VISPR$\downarrow$|UCF_Crime|HMDB51|THUMOS14|
> > |-|:-:|:-:|:-:|:-:|
> > |Raw Videos|64.3|83.0|62.4|60.3|
> > |Ours|45.6|82.2|62.0|60.1|
> >
> > ### Citations
> > [1] Hu, Edward J., et al. "Lora: Low-rank adaptation of large language models." ICLR 1.2 (2022): 3.
> >
> > [2] Li, Ming, et al. "Stprivacy: Spatio-temporal privacy-preserving action recognition." Proceedings of the IEEE/CVF International Conference on Computer Vision. 2023.
> >
> > [3] Ravfogel, Shauli, et al. "Null It Out: Guarding Protected Attributes by Iterative Nullspace Projection." Proceedings of the 58th Annual Meeting of the Association for Computational Linguistics. 2020.
> >
> > [4] Fioresi, Joseph, Ishan Rajendrakumar Dave, and Mubarak Shah. "Ted-spad: Temporal distinctiveness for self-supervised privacy-preservation for video anomaly detection." Proceedings of the IEEE/CVF international conference on computer vision. 2023.
> >
> > [5] Chen, Yingxian, et al. "Mgfn: Magnitude-contrastive glance-and-focus network for weakly-supervised video anomaly detection." Proceedings of the AAAI conference on artificial intelligence. Vol. 37. No. 1. 2023.

---

### Official Review · Reviewer_SgKC · 2025-11-01

**Soundness:** 3
**Presentation:** 3
**Contribution:** 3
**Rating:** 6
**Confidence:** 5

**Summary:**

This manuscript introduces SPLAVU, a latent-space anonymization framework that leaves the video encoder frozen and inserts a lightweight Anonymizing Adapter Module (AAM) to strip private attributes from clip features while preserving utility across tasks. The training couples (1) a clip-level, self-supervised privacy objective that maximizes NT-Xent on static clips to reduce shared spatial information, (2) co-training utility losses for action recognition, temporal action detection, and anomaly detection, and (3) a latent consistency loss to retain the encoder’s generalization on unseen tasks.
Evaluation measures privacy via VISPR cMAP with a linear attacker on static-clip representations and probes temporal sensitive attributes through Casia-B gait retrieval, while utility is tested on Kinetics-400, UCF101, HMDB51 (AR), THUMOS14 (TAD), and UCF-Crime (WSAD); the paper also proposes gender-bias protocols on NTU-Bias and Toyota Smarthome. Across backbones and tasks, SPLAVU reports 35% lower privacy leakage with ≤1–2% utility drop, and the self-attention AAM outperforms MLP adapters in privacy utility balance; in bias evaluations, it narrows gender-subclass gaps on Toyota Smarthome.

**Strengths:**

The manuscript proposes a latent‐space anonymization scheme (SPLAVU) that keeps the encoder frozen and adds a plug-and-play self-attention Anonymizing Adapter Module. This enables practical integration with low computational overhead and no backbone retraining.

The manuscript demonstrates a consistent privacy utility balance across multiple tasks (AR, TAD, WSAD) and different backbones, achieving sizable reductions in privacy leakage with ≤1–2% utility drop, and providing controllable trade-off curves.

The manuscript introduces a latent consistency loss that prevents overfitting the privacy budget and supports generalization to unseen tasks; ablations show that removing it degrades privacy and utility.

The manuscript evaluates temporally sensitive attributes and gender-bias protocols, showing mitigation of subclass gaps without sacrificing downstream performance.

**Weaknesses:**

The privacy evaluation is limited to VISPR with a linear attacker on static clips, which can overestimate the achieved protection and does not cover realistic threat scenarios (white/black-box, access to $f_E$ or $f_A$). Therefore, the manuscript should make the threat model explicit and test stronger attackers (e.g., MLP/ResNet or margin-based), also reporting ROC/PR curves to characterize risk more fully.

The choice to maximize NT-Xent as a privacy objective has limited theoretical support: it assumes shared information is “broken,” but it is unclear which attributes are actually removed and which persist. Therefore, the manuscript should expand the justification (e.g., connection to mutual information/subspace suppression) and add diagnostics showing which sensitive signals are effectively attenuated.

There is a modal mismatch between the problem (video) and the main privacy metric (attributes from static images); although a gait test is included, it remains a singular case. To better align the evaluation with the objective, it would be valuable to incorporate clip-level temporal attackers and to make the threat model explicit.

Qualitative visual results are missing, which would help readers interpret the practical effect of anonymization in video understanding. Therefore, the manuscript should include before/after examples, t-SNE projections of latents, activation maps, and representative failure cases to illustrate where privacy is gained and how utility is preserved.

Hyperparameter transparency and sensitivity of the composite objective are incomplete: the weights $\omega_{LC}$, $\omega_{T}$, and $\omega_{B}$ steer the privacy–utility balance, but their selection is not centralized. Therefore, the manuscript should consolidate a table with values per task/backbone and report sensitivity sweeps that guide configurations by use case

**Questions:**

Equation (7) shows a notation inconsistency: the objective uses $\omega_{LC}$, $\omega_{T}$, $\omega_{B}$, but the prose immediately refers to $\omega_{R}$ instead of $\omega_{LC}$.

Unify the symbol for the latent consistency weight across the manuscript (see lines 340--357).

In Equation (5) and its explanation, the text first suggests that minimizing increases similarity, but later states that the privacy loss is maximized; this back-and-forth can confuse readers.

Please clarify explicitly that the privacy term is maximized via the negative sign in (7), i.e., the total objective is minimized with $-\omega_{B} L_{B}$ (see lines 286--305 and 340--357).


In Criterion-1 (Equation 1), there is an extra argument “, $T_n$” inside $L_{T_n}(\cdot, T_n)$ on both sides of the approximate equality; this looks like a notation carryover.

Clean up the loss signature to avoid ambiguity (see lines 200--215).

In Appendix B.1 (Architecture), there is a typo: “Mulit-headAttention” should read “Multi-Head Attention.” Fix the spelling of the multi-head attention block (see lines 788--809).

In Section 5.2 (Bias), “complimentary protocol” should be “complementary protocol.” Adjust the term to the standard usage (see lines 386--399).

In Appendix A (NTU Bias protocol), “lets” should be “let’s.” Correct the contraction for grammatical accuracy (see lines 780--809).

Figure 1: the right panel (“Overall score / Task Accuracy (%) | Privacy (↓)”) is hard to read due to font size and density. Increase font sizes and clarify how the “overall score” is computed (refer to the corresponding formula; see lines 33--53 and 1026--1045).

Figure 2: label saturation (“AAM”, “MH Self-Attn Layer”, overlapping arrows). Simplify the legend (e.g., a symbol box) and scale typography for readability (see lines 162--215).

Variable naming standardization: Section 3.2 introduces $f_T^{*}$ and variants $f_{TAR}$, $f_{TTAD}$, $f_{TAD}$, while Algorithm 1 later uses $f_{\text{reco}}$, $f_{\text{tad}}$, $f_{\text{wsad}}$. Harmonize head names throughout text and code for traceability (see lines 232--269 and 1100--1133).

Definitions and units: in Section 4.2, metrics (Top-1, mAP, AUC) are not always paired with units/protocol specifics (e.g., clips per video, IoU thresholds, frame-level vs. segment-level). Add a brief table with metrics, units, and splits per dataset for clarity (see lines 324--377).

---

> ### Author Response · Authors · 2025-11-22
> **Response to Reviewer SgKC (1/2)**
>
> Thank you for your positive review and for offering thoughtful feedback that helps strengthen the paper.
>
> >"The privacy evaluation is limited to VISPR with a linear attacker on static clips, which can overestimate the achieved protection"
>
> Thank you for catching this - the description referring to the privacy classifier being a linear classifier was incorrect. We currently use a stronger MLP-based classifier (2 layers, ReLU activation, mapping from backbone output dimension to the number of classes). This setup achieves comparable private attribute prediction scores to the full ResNet encoder used in SPAct [1].
>
> >"[D]oes not cover realistic threat scenarios (white/black-box, access to $f_E$ or $f_A$"
>
> We describe and share results from white and black-box attack settings below. In the *black-box* setting, the attacker observes input–output pairs of the anonymizer and trains a separate inversion model where no gradients through $f_A$ are available. In the *white-box* setting, the attacker has full access to $f_A$ and instead performs gradient-based reconstruction by optimizing a synthetic embedding to match the observed anonymized output. In both cases, recovered embeddings are evaluated by feeding them into downstream models trained on non-anonymized features to measure action accuracy and privacy cMAP. See Table 1 for results. Notably, in the black-box setting, the inversion network successfully recovers action classification performance, yet does not improve attribute classification performance above random chance. The model is not robust to white box attacks, which are typically unrealistic in deployment scenarios. Robustness to white-box attacks can be a promising avenue for future work.
>
> **Table 1:** Feature inversion attack results. Our model demonstrates strong robustness to black-box attacks, where action performance is recovered but not private attribute prediction. Random chance on each dataset is included to contextualize performance.
>
> |Method|VP-HMDB51 Acc.$\uparrow$|VP-HMDB51 cMAP$\downarrow$|VP-UCF101 Acc.$\uparrow$|VP-UCF101 cMAP$\downarrow$|
> |-|:-:|:-:|:-:|:-:|
> |Raw Features|72.6|76.4|96.8|75.9|
> |Ours|72.1|70.5|96.8|69.6|
> |Random Chance|--|58.3|--|58.7|
> |Black Box Inversion|67.3|58.8|96.8|58.4|
> |White Box Inversion|68.0|73.7|94.5|72.6|
>
> >"There is a modal mismatch between the problem (video) and the main privacy metric (attributes from static images); although a gait test is included, it remains a singular case. To better align the evaluation with the objective, it would be valuable to incorporate clip-level temporal attackers and to make the threat model explicit."
>
> We adopted the VISPR image protocol due to established convention in the area. To better align the evaluation with the video modality, we additionally report results on VP-HMDB51 and VP-UCF101 [2], which provide video-level private-attribute annotations and thus serve as temporal privacy attackers. These results (Table 2) show consistent reductions in private-attribute prediction after anonymization, supporting that our method is effective beyond static-image protocols.
>
> **Table 2:** Privacy evaluation using video-level private attribute protocols VP-HMDB51 and VP-UCF101 [2].
> |Method|VP-HMDB51 Acc.$\uparrow$|VP-HMDB51 cMAP$\downarrow$|VP-UCF101 Acc.$\uparrow$|VP-UCF101 cMAP$\downarrow$|
> |-|:-:|:-:|:-:|:-:|
> |Raw Features|72.6|76.4|96.8|75.9|
> |Latent Anon. w/ Attribute Classifier|72.4|70.4|96.5|69.5|
> |Ours (w/ SSL/NT-Xent)|72.1|70.5|96.8|69.6|
>
>
> >"The choice to maximize NT-Xent as a privacy objective has limited theoretical support: it assumes shared information is “broken,” but it is unclear which attributes are actually removed and which persist. Therefore, the manuscript should expand the justification (e.g., connection to mutual information/subspace suppression) and add diagnostics showing which sensitive signals are effectively attenuated."
>
> Our results in Table 2 below show that maximizing the NT-Xent objective yields privacy removal comparable to an explicit adversarial approach that uses ground-truth private labels. This suggests that the contrastive objective is sufficient to suppress sensitive factors without requiring attribute supervision.

---

> ### Author Response · Authors · 2025-11-22
> **Response to Reviewer SgKC (2/2)**
>
> >Hyperparameter transparency and sensitivity of the composite objective are incomplete: the weights $\omega_{LC}$, $\omega_T$, and $\omega_B$ steer the privacy–utility balance, but their selection is not centralized. Therefore, the manuscript should consolidate a table with values per task/backbone and report sensitivity sweeps that guide configurations by use case.
>
> We keep the method simple by assigning equal weights to all task and privacy objectives ($\omega_T=1$ if used, else $\omega_T=0$). We would like to highlight that SOTA performance is still achieved this way. Table 9 (Appendix Section C) provides an ablation on the latent-consistency $\omega_{LC}$ weightage and Figure 10 provides a privacy–utility trade-off curve by varying the relative weighting of $\omega_B$ and $\omega_T$, illustrating how users can adjust the balance for their use case.
>
> **Notation inconsistency/readability:** Thank you for the detailed readability comments, we will update the manuscript with implemented fixes.
>
> ### Citations
> [1] Dave, Ishan Rajendrakumar, Chen Chen, and Mubarak Shah. "Spact: Self-supervised privacy preservation for action recognition." Proceedings of the IEEE/CVF Conference on Computer Vision and Pattern Recognition. 2022.
>
> [2] Li, Ming, et al. "Stprivacy: Spatio-temporal privacy-preserving action recognition." Proceedings of the IEEE/CVF International Conference on Computer Vision. 2023.

---

### Author Response · Authors · 2025-12-04
**AC Summary**

**Dear reviewers,**

We appreciate everyone for reading our paper carefully, providing detailed reviews, and for engaging well before the deadline. We hope that our responses have addressed any remaining concerns.

**Dear AC,**

Thank you for handling our submission under the current circumstances. To support your assessment, we summarize the key reviewer concerns and our corresponding clarifications and additions.

---

### Key Concern 1: Privacy Evaluation Scope (SgKC, u3Uc, dNWy)

**Concerns:**
- VISPR-only evaluation seemed limited; reviewers requested temporal attackers.

**Our Actions:**
- Added **video-level privacy evaluations** on **VP-HMDB51** and **VP-UCF101**, showing consistent leakage reduction.
    - Our model achieves performance comparable to one supervised with private attribute labels.

---

### Key Concern 2: Reconstruction Attack Feasibility (mpGY, dNWy)
**Concerns:**
- Anonymization might still permit reconstruction if a strong decoder exists.
- Latent-consistency loss might enable feature inversion.

**Our Actions:**
- Added a **VideoMAE decoder experiment**:
  - Applying the anonymizer to patch tokens prevents reconstruction.
  - Reconstruction fails from CLS tokens (raw or anonymized).
  - Qualitative results (Figure 4, Appendix Section C) and MSE metrics confirm strong resistance.
- Added **black-box and white-box inversion attack results**:
  - Black-box: recovers *utility* but *not private attributes* (remains at random chance).
  - White-box: recovers utility and and some private attribute prediction performance.
- Verified that latent-consistency loss *does not enable inversion*, supported by above black-box attack performance. Instead, it regularizes the anonymization, preventing the anonymizer from drifting into task-specific or degenerate feature spaces without overriding the privacy objective.

---

### Other Concerns

**Limited general task coverage:** Added **retrieval results** showing transfer to an unseen dataset/task (UCF101 retrieval using a K400-trained anonymizer).

**Gender bias results lacked comparison with existing debiasing methods:** Added comparison to debiasing baselines, showing comparable performance.

**Latent consistency ablation only shown on HMDB51 dataset:** Expanded latent consistency ablation to other datasets/tasks.

**Hyperparameter selection efficacy:** Emphasized that the main results use a *simple equal-weighting* scheme yet still achieve state-of-the-art performance.

**Missing analysis on effect of clip length and sampling strategy on privacy/utility:** Added clip-length and sampling strategy ablations.

**Notation/readability issues:** Updated manuscript to address raised concerns.

---

Across all reviews, the major concerns centered on privacy evaluation completeness and robustness of the anonymization mechanism. We responded with additional experiments (privacy attacks, reconstruction tests, ablations, retrieval transfer, etc.), expanded methodological clarification, and corrected notation.

Thank you for your time, we hope this summary is of use to your reviewing process.

---

### Meta-Review · Area_Chair_KXja · 2025-12-18

**Summary:**

# Decision

The authors propose a modular anonymization module to tackle visual privacy for video tasks, which performs in the latent space (extracted video features).

The reviewers acknowledged the applicability and usefulness of such a module, as well as its overall novelty. They also appreciated the thorough evaluation on downstream tasks, though they also noted the limited privacy evaluation (in image rather than video domain) and limited variability across the selected downstream tasks. They also expressed concerns w.r.t. lack of justification regarding some design choices and their impacts, as well as the lack of discussion regarding potential attacks.

Overall, the authors did cover most of the above concerns, e.g., by performing additional experiments (privacy evaluation with video-level private attribute protocols, evaluation of feature-inversion attacks, additional ablation studies, evaluation on new retrieval task, etc.), reflected in the updated manuscript.

We believe that this updated submission could benefit the community, hence our recommendation for approval.

------------
# Consolidated Reviews

## Strengths

### Relevant and modular solution for privacy in video applications
- Plug-and-play / modular solution [`SgKC`, `u3Uc`, `dNWy`]
- Consistent reductions in privacy leakage and privacy utility preservation [`SgKC`]
- Contribution: latent consistency loss to prevent overfitting privacy budget, generalizable to unseen tasks [`SgKC`]
- Contribution: idea of anonymizing latent features instead of input pixels or frames [`dNWy`]
    - *AC note: might be true for video-based anonymization, not image-based*

### Thorough evaluation on downstream tasks
- Thorough evaluation of utility preservation, performed on multiple downstream tasks and datasets [`u3Uc`, `mpGY`, `dNWy`]
- Evaluation of sensitive attributes and gender bias [`SgKC`, `u3Uc`, `dNWy`]

### Clarity
- Clear and well-motivated paper [`mpGY`, `dNWy`]

## Weaknesses

### Mismatch between claims and evaluations (video privacy, generalization to new tasks)
- Evaluation of privacy preservation limited to VISPR with linear attacker on static clips [`SgKC`, `u3Uc`]
- Mismatch between the problem (video) and the main privacy metric (attributes from static images) [`SgKC`, `u3Uc`]
- Generalizability-to-unseen-tasks claim not supported enough c.f. similarity between selected tasks  [`dNWy`]
- No systematic comparison with existing debiasing methods (c.f. gender-bias mitigation claims) [`u3Uc`]
- No evaluation on the impact of video-clip length c.f. impact on temporal consistency [`dNWy`]

### Limited discussion/justifications of key choices
- Lack of justification w.r.t. various design choices (e.g., architecture) and hyper-parameter values [`SgKC`, `u3Uc`]
- Lack of discussion/evaluation w.r.t. overhead/requirements for the utility loss $\mathcal{L}_t$ [`mpGY`]
- Lack of justification w.r.t. using NT-Xent as a privacy objective [`SgKC`]
- Risks of feature inversion or reconstruction attacks not considered/discussed [`dNWy`]
- Risk of training instability c.f. competing privacy vs. utility losses [`dNWy`]
- Unclear protocol for privacy evaluation [`dNWy`]
- Few unclear notations [`dNWy`]

### Ambiguity inherent to anonymization in latent space
- Challenges in interpreting results c.f. latent nature of anonymization [`mpGY`]
- Missing qualitative visualizations [`SgKC`, `dNWy`]

**Reviewer Concerns:**

See above for summary of main concerns shared by reviewers.

Overall, the authors have provided extensive justifications for some of their design choices (e.g., architecture, equal-weighting of losses, etc), and more importantly provided various new quantitative results to answer the reviewer concerns (privacy evaluation with video-level private attribute protocols, evaluation of feature-inversion attacks, additional ablation studies, evaluation on new retrieval task, etc.) ; thus covering most of the reviewers' concerns about the methodology and quantitative evaluation.

Some questions w.r.t. qualitative evaluation might remain after the authors shared some reconstruction results. Additionally, some concerns regarding the generalization to downstream video tasks might also remain, as the authors acknowledged that their work only applies to global-embedding tasks (e.g., action recognition, retrieval, etc.) and not to dense tasks (e.g., segmentation, captioning) -- this is now clarified in the submission, in the discussed Limitations.

**Reviewer Scores:**

### Reviewer `SgKC`
- **Original score:** 6
- **Score change:** more likely to keep their score, but might increase to 8. The authors covered most of their concerns, though some doubts might remain, e.g., regarding the provided qualitative results.

### Reviewer `u3Uc`
- **Original score:** 4
- **Score change:** likely to have kept their score / been unresponsive (c.f. shallow-ish review). The authors covered the reviewer's mostly superficial comments, but the reviewer might have been influenced by the other deeper reviews.

### Reviewer `mpGY`
- **Original score:** 6
- **Score change:** might have increased a bit. The reviewer was responsive to the authors, who clarified some misunderstanding.


### Reviewer `dNWy`
- **Original score:** 4
- **Score change:** might have remained borderline, though possibly leaning to ~6. The reviewer appreciated the authors' response but raised additional concerns. These concerns were partly covered in a second response by the authors.

---

### Decision · Program_Chairs · 2026-01-26

Accept (Poster)